# Multi-Reward as Condition for Instruction-based Image Editing

**Xin Gu**[1,2‡] **Ming Li**[1] **Libo Zhang**[2,3] **Fan Chen**[1] **Longyin Wen**[1] **Tiejian Luo**[2] **Sijie Zhu**[1,*]

[1]ByteDance Inc. [2]University of Chinese Academy of Sciences
[3]Institute of Software Chinese Academy of Sciences
`guxin21@mails.ucas.edu.cn`

## Abstract

High-quality training triplets (instruction, original image, edited image) are essential for instruction-based image editing. Predominant training datasets (e.g., InsPix2Pix) are created using text-to-image generative models (e.g., Stable Diffusion, DALL-E) which are not trained for image editing. Accordingly, these datasets suffer from inaccurate instruction following, poor detail preserving, and generation artifacts. In this paper, we propose to address the training data quality issue with multi-perspective reward data instead of refining the ground-truth image quality. 1) we first design a quantitative metric system based on best-in-class LVLM (Large Vision Language Model), i.e., GPT-4o in our case, to evaluate the generation quality from 3 perspectives, namely, instruction following, detail preserving, and generation quality. For each perspective, we collected quantitative score in $0 \sim 5$ and text descriptive feedback on the specific failure points in ground-truth edited images, resulting in a high-quality editing reward dataset, i.e., RewardEdit20K. 2) We further proposed a novel training framework to seamlessly integrate the metric output, regarded as multi-reward, into editing models to learn from the imperfect training triplets. During training, the reward scores and text descriptions are encoded as embeddings and fed into both the latent space and the U-Net of the editing models as auxiliary conditions. During inference, we set these additional conditions to the highest score with no text description for failure points, to aim at the best generation outcome. 3) We also build a challenging evaluation benchmark with real-world images/photos and diverse editing instructions, named Real-Edit. Experiments indicate that our multi-reward conditioned model outperforms its no-reward counterpart on two popular editing pipelines, i.e., InsPix2Pix and SmartEdit. Code is released at `https://github.com/bytedance/Multi-Reward-Editing`.

## 1 Introduction

Text instruction-based image editing provides a natural way for general users to express their requests and customize their assets easily. Predominant state-of-the-art methods for instruction-based image editing (Brooks et al., 2023; Zhang et al., 2024b;c; Huang et al., 2024) follow a data-driven pipeline to finetune pre-trained diffusion models (Rombach et al., 2022; Ren et al., 2025; Peng et al., 2024; Ye et al., 2025) with editing data triplets, i.e., (instruction, original image, edited image). Creating a high-quality dataset of the above triplets is thus essential for successful model training.

Predominant state-of-the-art methods for instruction-based image editing (Brooks et al., 2023; Zhang et al., 2024b;c; Huang et al., 2024; Ho & Salimans, 2022) follow a data-driven pipeline to create the editing triplets, from which they build a dataset to fine-tune a pre-trained diffusion model (Rombach et al., 2022). The most widely used InsPix2Pix (Brooks et al., 2023) dataset is created with a pre-trained text-to-image Stable Diffusion (SD) model (Rombach et al., 2022), Prompt-to-Prompt (Hertz et al., 2022) and a fine-tuned GPT-3 (Brown, 2020). The dataset can easily scale up to 300k triplets

---

∗ Corresponding author, sijiezhu@bytedance.com
‡ This work was done during the first author's internship at ByteDance

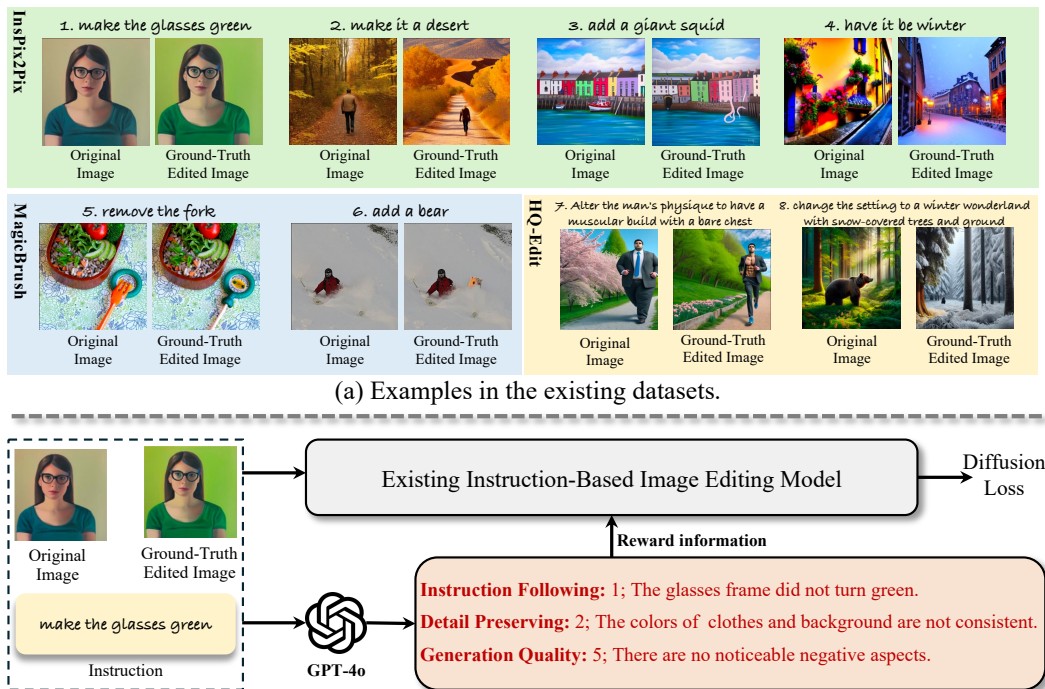

(a) Examples in the existing datasets.

(b) Our method adds multi-perspective reward as an input condition to guide editing model.

Figure 1: Existing image editing datasets and our method. Best viewed with zoom-in.

but the quality is unsatisfactory from three perspectives, i.e., *instruction following, detail preserving, and generation quality*. 1) *Instruction following* means that the model needs to closely and accurately follow the editing request, which we regard as the most important factor in instruction-based image editing. Since the SD model was originally trained for image generation tasks, it might fail to apply the correct editing action to the edited image. As shown in Fig. 1 (a), the text instruction is "make the glasses green" but the glasses in the ground-truth edited image are not green, which does not follow the major editing instruction. 2) *Detail Preserving* indicates how the model preserves identity, background or any other details that are not meant to be changed in the editing instruction. InsPix2Pix adopts prompt-to-prompt to generate edited images which could contain undesired modifications on the edited images. For example, the instruction of the first case in Fig. 1 (a) is to edit the color of the glasses, but the color of the clothes and background is also changed in the ground-truth edited image, which could lead to wrong supervision. 3) *Generation Quality* represents the relative quality of edited images compared to the input images, i.e., to determine whether the editing action introduces quality degradation like artifacts to the real-world input images. It is common for SD models to generate artifacts, especially for images with human or small objects. In the third case of Fig. 1 (a), the generated "giant squid" in the ground-truth image has serious artifacts (viewed with zoom-in).

MagicBrush (Zhang et al., 2024b) leverages a more powerful text-to-image model (i.e., DALL-E 2) and human workers to improve the training data quality on a relatively small scale. The background preserving is significantly improved due to mask-based editing. However, for the edited regions inside the mask, the edited image may contain undesired modification or generation artifacts due to occlusion or small objects (see example 5,6 in Fig. 1). HQEdit (Hui et al., 2024) adopts GPT-4V (gpt, b) and DALL-E 3 (dal) to improve the instruction and generation quality. However, the edited images are usually significantly modified on the regions that are not included in the editing instruction, leading to poor detail preserving on background or identity (see example 7,8 in Fig. 1). Hive (Zhang et al., 2024c) follows the same procedure of InsPix2Pix to create training data triplets, thus having a similar quality as InsPix2Pix. A relatively small-scale human feedback dataset is collected to improve the overall quality of the editing model, but it does not have detailed feedback information for the three perspectives of editing (i.e., following, preserving, and quality). *In a nutshell, the majority of training samples in existing datasets remain noisy which could lead to inaccurate supervision.*

In this paper, we propose to rectify the inaccurate supervision from a different perspective, i.e., introducing multi-perspective reward as an auxiliary input condition. 1) Instead of directly refining the quality of ground-truth edited images, we evaluate the training data triplets from three perspectives (i.e., instruction following, detail preserving, generation quality) with GPT-4o (gpt, a) to generate scores on a of $0 \sim 5$ and text description for unsatisfactory points. With proper prompt engineering, the generated reward/feedback is mostly aligned with humans. We collect 20k multi-perspective reward data in total for training, namely RewardEdit-20K. Examples of the scores and text description reward are included in Fig. 1 (b). 2) To integrate reward information into the existing instruction-based image editing framework, we first encode the reward score and reward text description separately as embeddings, and then concatenate them to obtain the reward condition. This reward condition is then integrated into the latent noise through an attention mechanism. To further enhance the guidance provided by the reward information, we also feed the reward condition into the U-Net (Ronneberger et al., 2015) of the SD model. 3) To evaluate the editing models on real-world photos and diverse instructions covering major 7 categories (defined in Sec. 5), we create an evaluation set with 80 high-quality Unsplash (uns) photos and 560 challenging instructions, which are initially generated by GPT-4o and verified by human annotators. We evaluate the model output from the three perspectives with GPT-4o in terms of yes/no accuracy and score from $0 \sim 5$. We also conduct a human evaluation with $0 \sim 5$ score from three perspectives to further verify the results. Experiments show that the proposed method can be combined with InsPix2Pix and SmartEdit with significant performance improvement.

We summarize the contributions as follows: ♠ The RewardEdit-20K dataset with multi-perspective reward data to address the limitations of existing image editing datasets. ♥ A novel framework to effectively integrate multi-perspective reward information as an additional condition to guide image editing. ♦ A real-world image editing evaluation benchmark Real-Edit and introduced a GPT-4o-based image editing evaluation method. ♣ Extensive experiments showing that the proposed method can be combined with existing editing models with a significant performance boost on all three perspectives, achieving state-of-the-art performance for both GPT-4o and human evaluation.

## 2 RELATED WORK

### 2.1 INSTRUCTION-BASED IMAGE EDITING

Recent instruction-based image editing methods (Zhang et al., 2024b; Peng et al., 2025; Zhang et al., 2024c; Guo et al., 2024; Gu et al., 2024) rely primarily on pre-trained text-to-image diffusion models. These methods leverage the powerful generative capabilities of these models and their understanding of textual descriptions to perform image editing. InsPix2Pix (Brooks et al., 2023), as a pioneering work, constructed a large-scale image editing dataset and successfully used instructions to edit images based on the stable diffusion model. MagicBrush (Zhang et al., 2024b) addressed the issue of unrealistic images in InsPix2Pix by creating a manually annotated dataset to achieve realistic image editing. SmartEdit (Huang et al., 2024) addressed the limitation of InstructPix2Pix in handling only simple instructions by employing LLava (Liu et al., 2024) to comprehend complex instructions. HIVE (Zhang et al., 2024c) proposed to utilize human feedback to optimize image editing models, aligning them with human preferences. *However, the major training data still has a similar quality as the InsPix2Pix dataset, and the noisy supervision problem remains unaddressed.*

### 2.2 REWARD MECHANISM FOR DIFFUSION MODELS

Inspired by the success of reward fine-tuning in large language models (Rafailov et al., 2024; Han et al., 2024; Zhang et al., 2024a), a series of works have attempted to directly optimize reward model scores (Xu et al., 2024; Fan et al., 2024) or human preference rankings (Wallace et al., 2024; Liang et al., 2024) to align text-to-image diffusion models, thus improving the quality, aesthetics, and text-image alignment of the generated images. For text-to-image, Pony Diffusion employs a CLIP-based aesthetic ranking method to generate reward scores to improve the quality of generated images. For image editing, ByteEdit (Ren et al., 2024) customizes a reward model specifically for inpainting and outpainting editing tasks to identify the consistency of images beyond the mask area before and after

editing. HIVE (Zhang et al., 2024c) trains a reward model to generate a single reward score for each edited image. The scores are then combined with text instructions and encoded via CLIP (Radford et al., 2021a) to improve editing performance. *However, there are multiple perspectives to determine the quality of an edited image given an input image and instruction, which cannot be covered by one single reward score.* Also, adding the reward scores into the text instruction does not fully exploit the reward information, as the CLIP text encoder is not sensitive to numbers. *It remains challenging to effectively integrate multi-perspective reward information into existing image editing frameworks.*

## 3   REWARDEDIT-20K: A MULTI-REWARD DATASET FOR IMAGE EDITING

**Collection Process.** In this section, we discuss our procedure for collecting the RewardEdit-20K dataset. First, we randomly selected 20K training triplets from the InsPix2Pix dataset, where each triplet contains an original image, an edited image, and an editing instruction. Then, we used GPT-4o, setting up three types of prompts based on instruction following, detail preserving, and generation quality. GPT-4o was asked to perform evaluations on these three aspects for each triplet. Finally, we obtained 20K reward data consisting of reward scores and reward texts. The reward collection process is illustrated in Fig. 2. Due to limited space, we only show the core prompts in the figure, while the complete prompts are provided in the appendix.

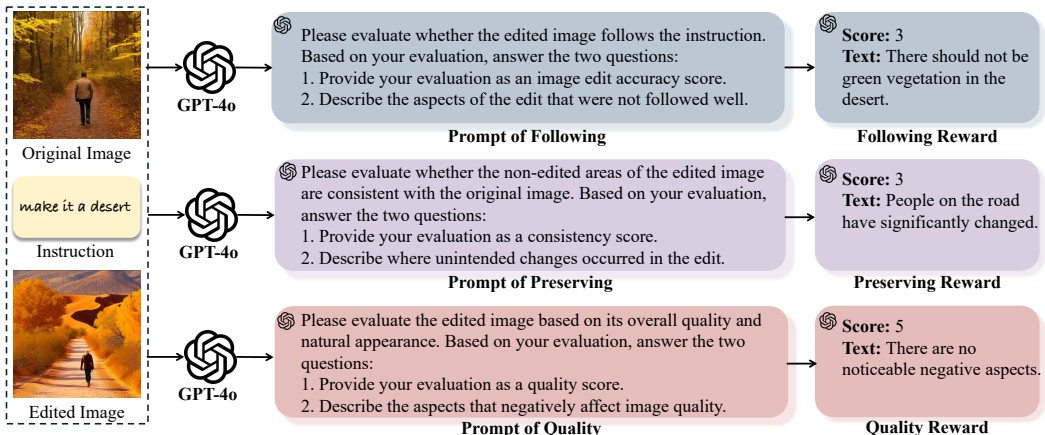

Figure 2: Generation process of reward data. Given the editing triplets, reward data was generated using GPT-4o by setting prompts from different perspectives.

**Data statistics.** We summarize the statistics of the reward data. Fig.3 shows the distribution of the reward scores, revealing that in all three aspects, samples with scores less than 5 exceed 50%, indirectly indicating that the majority of training samples in the InsPix2Pix dataset remain noisy. Fig.4 uses word clouds to display the most frequent words in the reward texts. These words reflect the main issues present in the original dataset. For example, the high frequency of 'executed' and 'poorly' in the instruction-following aspect indicates failures in following instructions, 'unintended' and 'change' in the detail-preserving aspect reflect inconsistencies in non-edit areas, and 'lighting' and 'shadow' in the quality aspect highlight quality issues in the edited images.

## 4   METHODOLOGY

**Overview.** In this section, we first introduce the most general image editing framework (Sec.4.1). Then, we present our framework that uses multi-reward as an input condition (Sec.4.2). Finally, we offer a detailed explanation of the multi-reward condition module (Sec. 4.3).

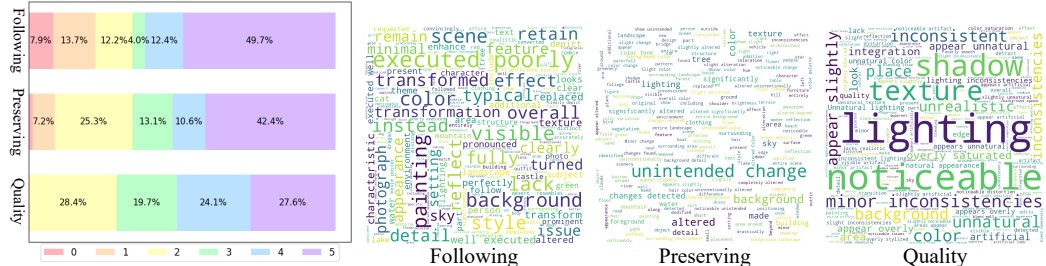

Figure 3: Distribution of reward score.

Figure 4: Word cloud of reward text.

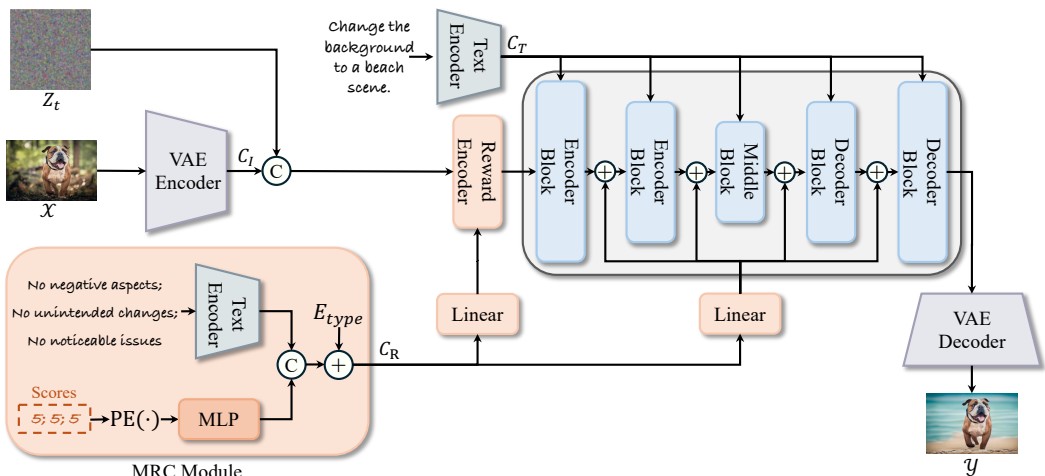

Figure 5: The overall framework of our approach. The original image $x$ is first encoded into an image condition by the VAE encoder. This image condition $c_I$ is then concatenated with latent noise $Z_t$ to serve as the query for the reward encoder, with the reward condition $c_R$ as the key/value. The resulting latent noise, containing reward information, is used as the input for the U-Net module. Meanwhile, the instruction is encoded into a text condition $c_T$ by the text encoder, which is fed into each block of the U-Net. To further enhance reward guidance, we incorporate the reward condition after each block. Finally, the U-Net's output is decoded by the VAE decoder into the edited image $y$.

## 4.1 PRELIMINARY: GENERAL IMAGE EDITING FRAMEWORK

InsPix2Pix (Brooks et al., 2023), as one of the pioneering works in the field of instruction-based image editing, can edit images according to the given instructions. Specifically, given the original image $x$, the text instruction $t$, and the edited image $y$, first use the VAE encoder to extract the encoded latent $z$ and original image conditioning $c_I$, that is, $z = \mathcal{E}(y), c_I = \mathcal{E}(x)$. Similarly, use the text encoder to extract the text condition $c_T$. Through the diffusion process, noise is added to $z$ to generate latent noise $z_t$, where the noise level increases over timesteps $t \in T$. Then, train a network that predicts the noise added to the noisy latent $z_t$ given the original image conditioning $c_I$ and the text instruction conditioning $c_T$. The specific objective of latent diffusion is as follows:

$$\mathcal{L}_{\text{InsPix2Pix}} = \mathbb{E}_{z,c_I,c_T,\epsilon \sim \mathcal{N}(0,1),t}[\|\epsilon - \epsilon_\delta(t, \text{concat}[z_t, c_I], c_T))\|_2^2] \quad (1)$$

where $\epsilon$ is the unscaled noise, $t$ is the sampling timestep, $z_t$ is latent noise at step $t$. SmartEdit (Huang et al., 2024), the state-of-the-art instruction-based image editing model, uses the same architecture as InsPix2Pix but upgrades text encoder from CLIP (Radford et al., 2021b) to LLaVA (Liu et al., 2024).

Although methods like InsPix2Pix and SmartEdit have shown compelling results in image editing, they are still affected by noise present in the training data, thus limiting their performance. To address this, we propose using multi-perspective rewards as an additional condition to correct the bias introduced by the training data.

## 4.2 MULTI-REWARD AS INPUT CONDITION

We adopted an architecture similar to InsPix2Pix and SmartEdit. On this basis, to utilize rewards to guide the model, we designed a multi-reward condition (MRC) module to extract the reward condition and used a reward encoder to integrate the reward condition into the diffusion process. Additionally, to further enhance the guidance of reward information, we also incorporated the reward condition after each block in the U-Net module. The framework as shown in Fig. 5. Given the original image $x$, the edited image $y$ is generated under the guidance of the instruction text $t$ and the reward data. First, we use the VAE encoder to extract the latent representation $c_I$ of the original image. As in InsPix2Pix, concatenate $c_I$ with latent noise $Z_t$ and fuse them through convolution to obtain $Z'_t$. Then, we use the proposed MRC module to generate a reward condition $c_R$ (Details in Sec. 4.3). To utilize the reward condition $c_R$ to guide image editing, we integrate the reward condition $c_R$ into the encoded latent noise through a reward encoder, which consists of 1 standard transformer encoder block (Vaswani, 2017). Specifically, let latent noise $Z'_t$ serve as query and reward condition as key/value, this process can be expressed as follows,

$$Z''_t = \text{CA}(Z'_t, \text{Linear}_1(c_R)) \tag{2}$$

where $\text{CA}(\mathbf{z}, \mathbf{u})$ denotes the transformer encoder block with $\mathbf{z}$ generating query and $\mathbf{u}$ is the key/value. $\text{Linear}_1(\cdot)$ denotes linear projection, which aligns the dimension of the reward condition with the latent noise. To further enhance the guidance of the reward information, we also add the reward condition after each block in the U-Net module. The input to the $i$ th block in U-Net is as follows,

$$\hat{z}_i = \text{UB}_{i-1}(\hat{z}_{i-1}) + \text{Linear}_2(c_R) \tag{3}$$

where $\text{UB}_{i-1}(\cdot)$ denotes $i$-1 th the blocks in U-Net. $\text{Linear}_2(\cdot)$ aligns the dimension of the reward condition with the U-Net. After that, the output of the U-Net module is fed into the VAE decoder to generate the edited image $y$. The specific process can be formulated as:

$$\mathcal{L}_{\text{Reward}} = \mathbb{E}_{z,c_I,c_T,c_R,\epsilon\sim\mathcal{N}(0,1),t}[\|\epsilon - \epsilon_\delta(t, \text{concat}[z_t, c_I], c_T, c_R))\|_2^2] \tag{4}$$

## 4.3 MULTI-REWARD CONDITION MODULE

We use an additional reward condition $c_R$ from the MRC module to guide the model in generating the desired edited image. The MRC module is responsible for generating the reward condition from the reward text and reward score. For the reward text, we use the text encoder in Stable Diffusion model to extract text embeddings $E_t$, as follows:

$$E_t = \text{Encoder}_{\text{text}}(\text{Concat}[\mathcal{T}_f, \mathcal{T}_p, \mathcal{T}_q]) \tag{5}$$

where $\mathcal{T}_f, \mathcal{T}_p, \mathcal{T}_q$ are the reward text in terms of following, preserving, and quality, respectively. For the reward scores, we use absolute positional encoding (Vaswani, 2017), which utilizes sine and cosine functions to convert the scores into vectors, and then extract embeddings $E_s$ with an MLP module. The process mentioned above is represented as:

$$E_s = \text{MLP}(\text{Concat}[\text{PE}(\mathcal{S}_f), \text{PE}(\mathcal{S}_p), \text{PE}(\mathcal{S}_q)]) \tag{6}$$

where $\mathcal{S}_f, \mathcal{S}_p, \mathcal{S}_q$ are the reward scores in terms of following, preserving, and quality, respectively. $\text{MLP}(\cdot)$ and $\text{PE}(\cdot)$ denote the MLP module and position encoding. Finally, concatenate the text embedding and the score embedding, and add the type embedding to obtain the reward condition $c_R$.

## 5 EVALUATION BENCHMARK AND METRICS

**Evaluation Data.** To more comprehensively evaluate the model's ability to edit real images based on instructions, we constructed a new image editing evaluation benchmark, Real-Edit, using real-world images. Compared to existing evaluation benchmarks, our proposed test set includes higher-quality images and a greater variety of editing instructions. We first carefully selected 80 high-quality images from the Unsplash website as the original images. The categories of these images are shown in Fig.6. Then, using GPT-4o, we generated 7 different editing instructions for each image based on its content, including local, remove, add, texture, background, global, and style edits, as shown in Fig.7.

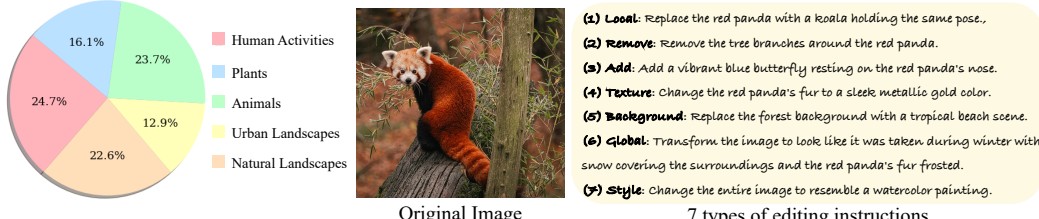

Figure 6: Distribution of different categories of images in Real-Edit.

Figure 7: An example in Real-Edit.

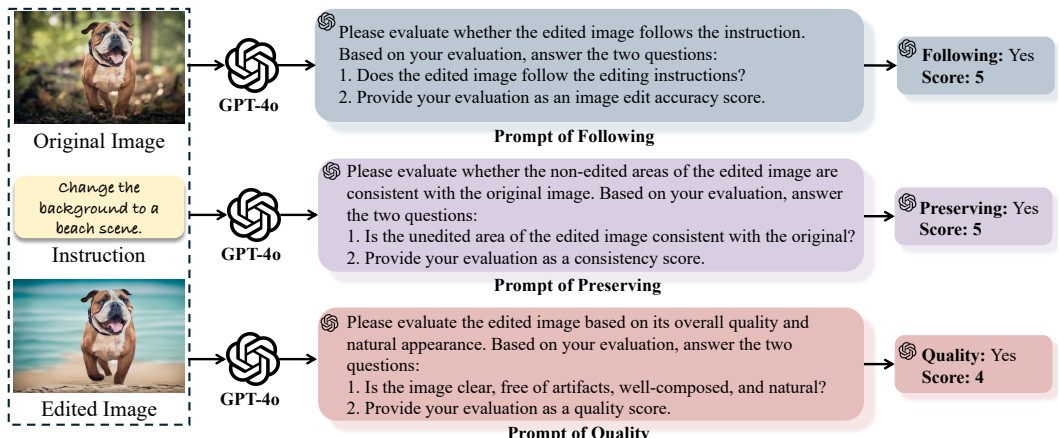

Figure 8: Evaluation process. The generated edited image, original image, and instruction are input into GPT. Three prompts are designed to evaluate from three different aspects. For each aspect, determine whether the criteria are met and assign a score (ranging from 0 to 5).

**Evaluation Metrics.** To more accurately evaluate the performance of the editing model, we used GPT-4o to evaluate the edited images based on the original images and instructions. The evaluation is conducted from three perspectives as follows: (1) Following: Determine whether the edited image has been modified according to the editing instructions. (2) Preserving: Evaluate whether the non-edited aspects of the original and edited images remain consistent. (3) Quality: Focuses on the overall quality of the edited image compared to the input image, including aspects such as clarity, composition, and lighting. The detailed evaluation process is illustrated in Fig. 8. Due to limited space, we only show the core prompts in the figure, while the complete prompts are provided in the appendix. In the later quantitative results, we have supplemented each edited image with evaluation scores.

## 6 EXPERIMENTS

### 6.1 IMPLEMENTATION DETAILS.

Our method is implemented in Python using PyTorch. The MRC module, reward encoder, and the connected linear layer are randomly initialized. All other modules are initialized from the pre-trained InsPix2Pix model (Brooks et al., 2023). During training, we only optimize the MRC module, the U-Net module, the reward encoder, and the connected linear layers. And we use the Adam (Kingma, 2014) optimizer with an initial learning rate of $5e-5$, a weight decay of $1e-2$, and a warm-up ratio of $0$. We resize the images to $256$ and apply random cropping during training and resize the shorter side to $512$ during inference.

### 6.2 STATE-OF-THE-ART COMPARISON

To validate the efficacy of our method, we compared it against other image editing methods. The results on Real-Edit are summarized in Tab. 1. Reward-InsPix2Pix, which is fine-tuned based on reward data, significantly improved all metrics compared to InsPix2Pix: following accuracy increased

Table 1: Comparison with existing state-of-the-art methods on Real-Edit.

| Method | Edit Data | Following | | Preserving | | Quality | |
|---|---|---|---|---|---|---|---|
| | | Acc | Score | Acc | Score | Acc | Score |
| KOSMOS-G (Pan et al., 2023) | 9M | 51% | 2.82 | 9% | 1.43 | 27% | 3.20 |
| MagicBrush (Zhang et al., 2024b) | 0.31M | 51% | 2.90 | 70% | 3.85 | 50% | 3.67 |
| MGIE (Fu et al., 2023) | 1M | 40% | 2.43 | 45% | 2.79 | 38% | 3.35 |
| InstructDiffusion (Geng et al., 2024) | 0.86M | 52% | 2.87 | 54% | 3.17 | 47% | 3.58 |
| HIVE (Zhang et al., 2024c) | 1.1M | 54% | 2.93 | 56% | 3.36 | 53% | 3.72 |
| HQ-Edit (Hui et al., 2024) | 0.5M | 51% | 2.84 | 16% | 1.63 | **54%** | **3.84** |
| InsPix2Pix (Brooks et al., 2023) | 0.3M | 52% | 2.94 | 53% | 3.31 | 50% | 3.69 |
| Reward-InsPix2Pix | 0.32M | 63% | 3.39 | 58% | 3.43 | **54%** | 3.80 |
| SmartEdit (Huang et al., 2024) | 1.17M | 64% | 3.50 | 66% | 3.70 | 45% | 3.56 |
| Reward-SmartEdit | 1.19M | **69%** | **3.72** | **74%** | **4.00** | 49% | 3.67 |

Table 2: Results on MagicBrush test set (%).

| Method | CLIP-I | CLIP-T |
|---|---|---|
| MagicBrush | 90.7 | **30.6** |
| MGIE | 90.9 | 30.5 |
| InstructDiffusion | 89.2 | 30.2 |
| InsPix2Pix | 85.4 | 29.2 |
| Reward-InsPix2Pix | 88.9 | 29.8 |
| SmartEdit | 90.4 | 30.3 |
| Reward-SmartEdit | **91.3** | 30.5 |

Table 3: Human evaluation scores for SmartEdit and Reward-SmartEdit on Real-Edit benchmark.

| Method | Following | Preserving | Quality |
|---|---|---|---|
| SmartEdit | 3.09 | 3.18 | 2.58 |
| Reward-SmartEdit | **3.34** | **3.46** | **2.73** |

by 11%, score by 0.45, preserving accuracy by 5%, score by 0.12, quality accuracy by 4%, and score by 0.11. SmartEdit, as the leading image editing model, achieved a new SOTA performance after fine-tuning based on reward data. The proposed MRC module needs to be trained separately for each model. Despite using much less additional editing data (0.02M) compared to KOSMOS-G (9M), our method significantly improved both InsPix2Pix and SmartEdit, demonstrating its efficiency.

We also evaluated our method on the common evaluation benchmark MagicBrush (Zhang et al., 2024b), as shown in Tab. 2. Using reward data for fine-tuning still improves the performance of the editing model. Specifically, it helps InsPix2Pix improve by 3.5 on CLIP-I and 0.6 on CLIP-T, and it helps SmartEdit improve by 0.9 on CLIP-I and 0.2 on CLIP-T. These results once again validate the effectiveness of our method.

## 6.3 HUMAN EVALUATION

To further validate the performance of our method against state-of-the-art methods, we conduct a human evaluation. Specifically, we selected the best-performing SmartEdit and the Reward-SmartEdit model fine-tuned using reward data. We collected the edited images they generated on Real-Edit, with 560 samples each. We then recruited 10 professional annotators to evaluate the edited images based on the three aforementioned aspects. The evaluation results are shown in Tab. 11. As indicated in the table, Reward-SmartEdit significantly outperformed the original SmartEdit, further demonstrating the effectiveness of our method. The human evaluation score is in general lower than GPT-4o scores on all methods (Fig. 13 in Appendix), but the rank of different methods are consistent. We guess that the reason for this discrepancy may be that human evaluators often have higher expectations and subjective perceptions, making them more critical of details and quality.

## 6.4 ABLATION STUDY

**Ablation for two types of reward data.** The reward data consists of reward scores and reward text. To explore the effects of these two types of reward information, we conducted ablation experiments on Real-Edit. As shown in Tab. 4, our baseline model without reward information (❶) achieves a following accuracy of 49%, preserving the accuracy of 38%, and quality accuracy of 32%. When the

reward score is applied alone, these metrics improve by 12%, 17%, and 21%, respectively (❶ *vs.* ❷). When the reward text is used alone, the metrics improve by 11%, 14%, and 19%, respectively (❶ *vs.* ❸). Combining both reward score and reward text yields the best results, with the following accuracy, preserving accuracy, and quality accuracy reaching 63%, 58%, and 54%, respectively (❹). These results clearly validate the efficacy of incorporating reward information.

Table 4: Ablation of two types of reward data on Real-Edit.

| | Reward | | Following | | Preserving | | Quality | |
|---|---|---|---|---|---|---|---|---|
| | Score | Text | Acc | Score | Acc | Score | Acc | Score |
| ❶ | | | 49% | 2.77 | 38% | 2.59 | 32% | 3.15 |
| ❷ | ✓ | | 61% | 3.29 | 55% | 3.40 | 53% | 3.70 |
| ❸ | | ✓ | 60% | 3.25 | 52% | 3.30 | 51% | 3.25 |
| ❹ | ✓ | ✓ | **63%** | **3.39** | **58%** | **3.43** | **54%** | **3.80** |

**Ablation for different methods of reward integration.** To utilize reward information to guide the image editing model, we integrate the reward condition into the edit model using two methods: attention in the reward encoder and addition in the U-Net. To explore the impact of these methods, we conducted ablation experiments, as shown in Tab. 5. When using attention alone, following, preserving, and quality accuracy improved by 11%, 8%, and 16%, respectively (❶ *vs.* ❷). Using addition alone, the metrics improved by 8%, 18%, and 20% (❶ *vs.* ❸). Combining both attention and addition achieved the best performance across all metrics (❹), validating the importance of these methods for integrating reward conditions.

Table 5: Ablation of methods for integrating reward information on Real-Edit.

| | Attention | Addition | Following | | Preserving | | Quality | |
|---|---|---|---|---|---|---|---|---|
| | | | Acc | Score | Acc | Score | Acc | Score |
| ❶ | | | 49% | 2.77 | 38% | 2.59 | 32% | 3.15 |
| ❷ | ✓ | | 60% | 3.27 | 46% | 3.11 | 48% | 3.64 |
| ❸ | | ✓ | 57% | 3.15 | 56% | **3.44** | 52% | 3.76 |
| ❹ | ✓ | ✓ | **63%** | **3.39** | **58%** | 3.43 | **54%** | **3.80** |

**Ablation for different reward scores during inference.** During inference, we set the reward scores for following, preserving, and quality to 5, and set the reward text to 'None'. To investigate whether the model's editing performance is influenced by reward information, we conducted experiments as shown in Tab.6. From the results in Tab.6, we observe that as the scores decrease, the model's editing performance in all three aspects significantly declines. Specifically, when the scores dropped from 5 to 0, the accuracy for following, preserving, and quality decreased by 9%, 28%, and 19%, respectively. This indicates that our model can understand the meanings of different scores and achieve a certain degree of controllable generation in the quality of the generated images.

Table 6: Ablation of different reward scores.

| | Reward Score | | | Following | | Preserving | | Quality | |
|---|---|---|---|---|---|---|---|---|---|
| | F | P | Q | Acc | Score | Acc | Score | Acc | Score |
| ❶ | 0 | 0 | 0 | 54% | 2.90 | 30% | 2.31 | 35% | 3.37 |
| ❷ | 3 | 3 | 3 | 59% | 3.19 | 42% | 2.94 | 47% | 3.64 |
| ❸ | 5 | 5 | 5 | **63%** | **3.39** | **58%** | **3.43** | **54%** | **3.80** |

## 6.5 QUALITATIVE COMPARISON

To further qualitatively validate the effectiveness of our proposed method, we presented the results of our method on InsPix2Pix and SmartEdit, as well as the results of other image editing methods, as shown in Fig. 9. In Fig. 9, both the Reward-InsPix2Pix and Reward-SmartEdit outperform the original InsPix2Pix and SmartEdit, and their editing performance is also better compared to other methods, showing the effectiveness of our method.

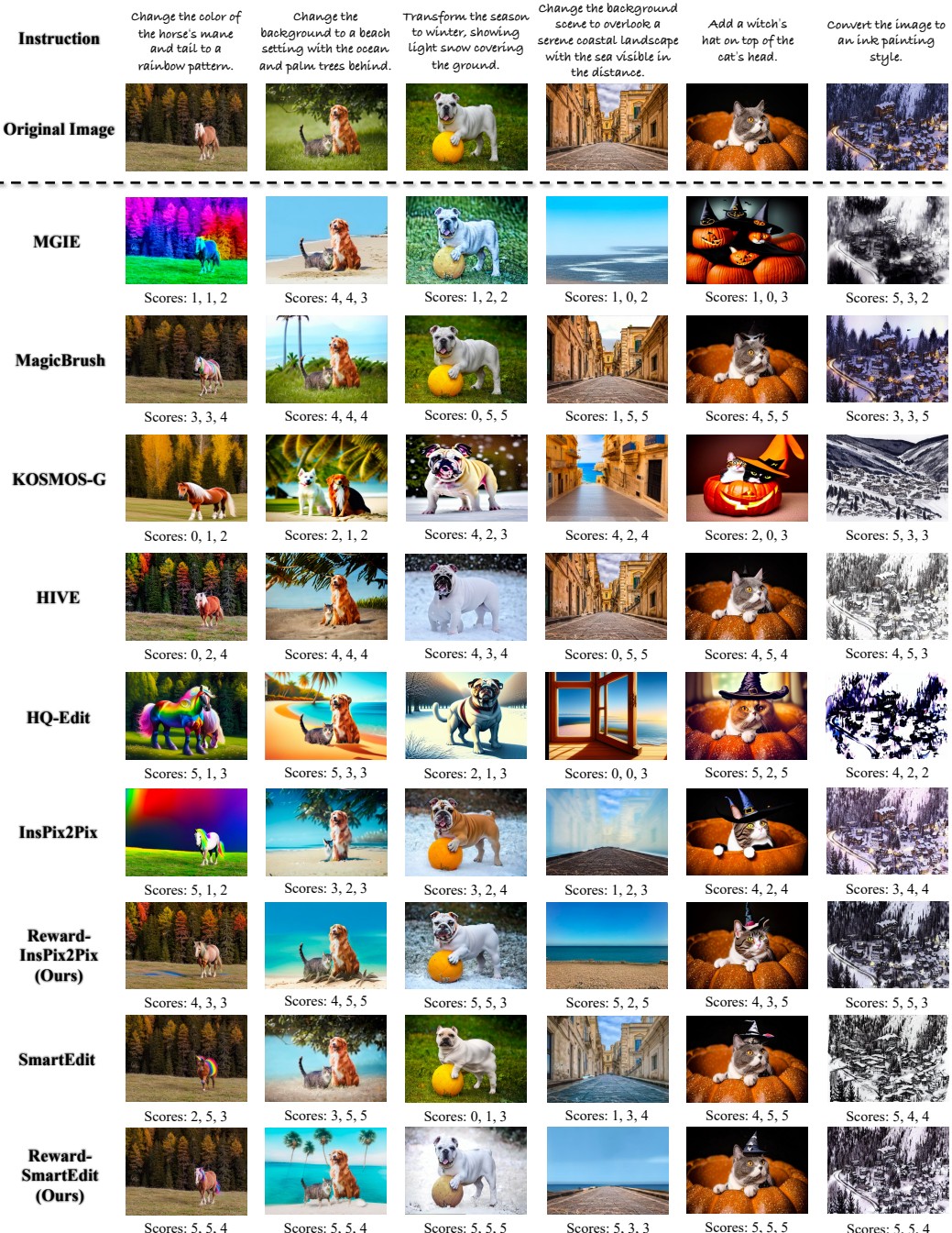

Figure 9: Qualitative results on Real-Edit. The three scores below the image are given by GPT-4o in three aspects: instruction following, detail preservation, and generation quality.

# 7 CONCLUSION

We propose a novel framework to rectify the noisy supervision for instruction-based image editing models by adding multi-perspective reward data as additional conditions. We collect 20k multi-perspective reward data, named RewardEdit-20k, using a subset of InxPix2Pix dataset and GPT-4o. Additionally, we presented the Real-Edit benchmark and a GPT-4o-based evaluation method. Extensive experiments show that our approach significantly enhances performance across all perspectives, achieving state-of-the-art results in both GPT-4o and human evaluations.

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

# APPENDIX

## A FAILURE CASE ANALYSIS

To explore the limitations of our method, we collected and analyzed failed cases. The analysis revealed two main limitations of our method. The first limitation is that during testing, even when the given multi-perspective reward scores are all 5, the generated edited image does not always achieve a score of 5. This indicates that the reward information does not always perfectly guide the model, especially in some complex cases. The second limitation is that our method has difficulty accurately understanding the quantifiers and spatial position words in the instructions, as shown in Fig. 10. This may be due to the model's insufficient understanding of fine-grained textual features. In future work, we will explore ways to improve the model's understanding of fine-grained semantics for image editing.

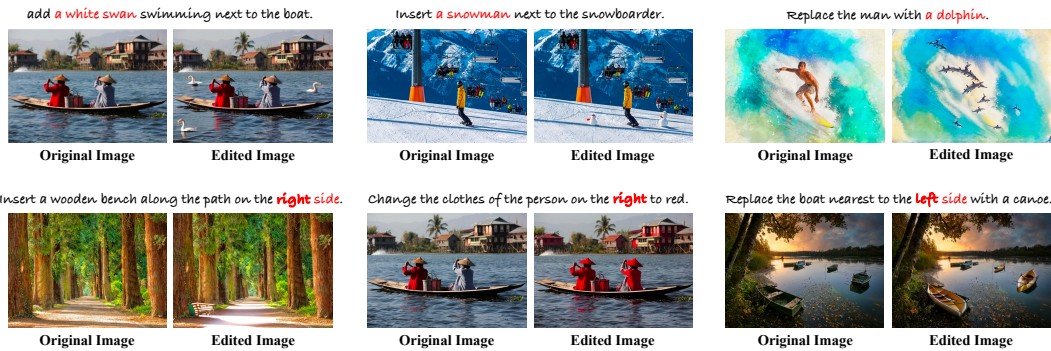

Figure 10: Examples of failure cases. In the first three edited images, the number of objects is incorrect, while in the last three edited images, the spatial positions of the objects are incorrect.

## B EXAMPLES IN REWARDEDIT-20K

We show examples from RewardEdit-20K, as shown in Fig. 11. For each triplet (instruction, original image, edited image), there are three perspectives of rewards: instruction following, detail preserving, and generation quality. Each reward consists of a score and text. The reward score reflects the overall quality, while the reward text provides more detailed information.

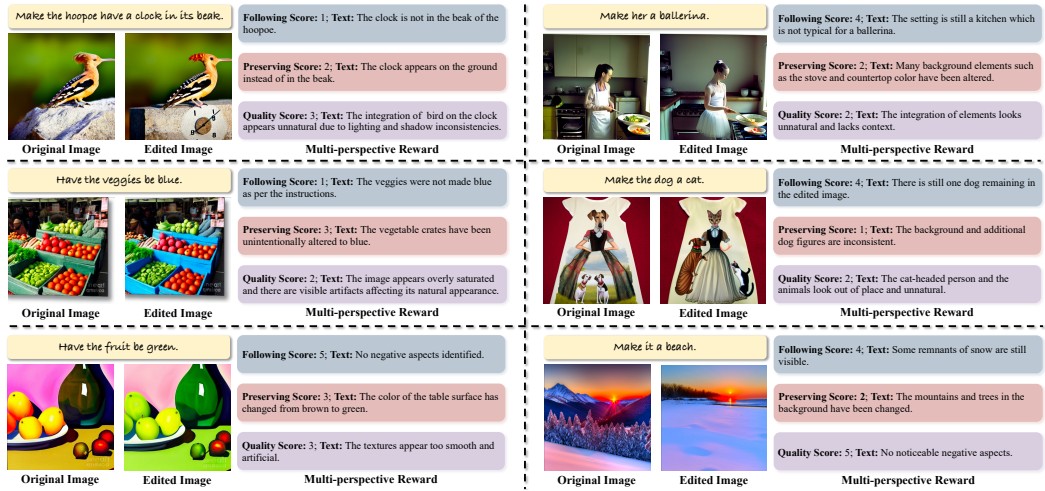

Figure 11: Examples from the RewardEdit-20K dataset. Best viewed with zoom-in.

## C ADDITIONAL EXPERIMENT RESULTS

### C.1 EVALUATION BASED ON EXISTING METRICS

We also evaluated Following, Preserving, and Quality based on existing evaluation metrics, as shown in Fig. 7. We recalculated the performance of existing methods and our method based on the CLIP score and the FID score. Specifically, the CLIP feature similarity between the edited image and the instruction is the Following score, the similarity between the original and edited images is the Preserving score, and the FID between the original and edited images is the Quality score. The table shows that our method still achieved promising results and improvements over the baseline. However, these metrics also have limitations: 1) When the editing instruction and the images are complicated, CLIP/FID score can not accurately represent the following/preserving/quality of the edited image, e.g., CLIP can not distinguish left/right. 2) the range of the following score and preserving score is relatively small, which may make it hard to distinguish performance differences between methods.

Table 7: Comparison of different methods based on existing evaluation metrics.

| Method | Following (CLIP) | Preserving (CLIP) | Quality (FID) ↓ |
| --- | --- | --- | --- |
| KOSMOS-G | 26.8 | 86.4 | 3.01 |
| MagicBrush | 25.2 | 91.9 | 2.86 |
| MGIE | 26.4 | 87.0 | 3.09 |
| InstructDiffusion | 26.3 | 86.4 | 2.89 |
| HIVE | 26.3 | 89.0 | 3.08 |
| HQ-Edit | 28.5 | 77.2 | 3.59 |
| InsPix2Pix | 27.0 | 82.3 | 3.51 |
| Reward-InsPix2Pix | 27.5 | 83.8 | 3.31 |
| SmartEdit | 26.5 | 87.7 | 2.80 |
| Reward-SmartEdit | 26.9 | 90.0 | 2.77 |

### C.2 ABLATION STUDY OF EDITING DATA

The 20K samples in our RewardEdit-20K dataset are randomly sampled from InsPix2Pix. Our motivation is that constructing a perfect image editing dataset is challenging, and the ground truth in existing image editing datasets often contains issues. Therefore, we propose using multi-perspective rewards to rectify the inaccurate supervision. To more fairly demonstrate the role of multi-perspective rewards, we conducted the ablation experiments shown in Tab. 8. The experimental results indicate that, with the same data, using multi-perspective rewards significantly improves performance compared to the baseline, demonstrating the effectiveness of multi-perspective rewards.

Table 8: Ablation study of editing data.

| Method | Edit Data | Following | Preserving | Quality |
| --- | --- | --- | --- | --- |
| Baseline | 0.30M | 2.77 | 2.59 | 3.15 |
| | 0.32M | 2.90 | 2.88 | 3.52 |
| Ours | 0.32M | 3.39 | 3.43 | 3.80 |

### C.3 ABLATION STUDY ON CHALLENGING SAMPLES

To investigate whether our reward model can generate better edited images for challenging editing samples in InsPix2Pix, we first randomly selected 500 samples from RewardEdit-20K with scores not exceeding 2. Then, we used our reward model to generate edited images based on the original images and instructions of these samples, and scored them using GPT-4o. The experimental results are shown in Tab. 9. "Original" represents the average scores of the original edited images of these samples across three metrics, while "Ours" represents the scores of the edited images generated by

our reward model. From the table, it can be observed that the edited images generated by our method significantly outperform the original edited images on all three metrics, indicating that our method can generate better results for these difficult cases.

Table 9: Comparison of edited images for challenging samples in InsPix2Pix.

| Method | Following | Preserving | Quality |
|---|---|---|---|
| Original | 1.15 | 1.69 | 1.99 |
| Ours | 2.92 | 4.10 | 3.68 |

### C.4 ABLATION STUDY OF EACH PERSPECTIVE REWARD

We find that analyzing the impact of each perspective reward is beneficial, and we conduct additional experiments by training on each perspective separately. As shown in Tab. 10, the following score reached 3.40 with only the instruction following reward, the preserving score reached 3.54 with only the detail preserving reward, and the quality score reached 3.95 with only the generation quality reward. These results demonstrate the effectiveness of each perspective reward.

Table 10: Ablation of each perspective reward. 'IF', 'DP' and 'GQ' are instruction following, detail preserving and generation quality reward.

| IF | DP | GQ | Following | Preserving | Quality |
|---|---|---|---|---|---|
| ✓ | | | 3.40 | 3.25 | 3.72 |
| | ✓ | | 3.23 | 3.54 | 4.00 |
| | | ✓ | 3.20 | 3.23 | 3.95 |
| ✓ | ✓ | ✓ | 3.39 | 3.43 | 3.80 |

### C.5 ABLATION STUDY OF TRAINING RESOLUTION

We chose to train at a resolution of 256 to maintain consistency with other methods (InsPix2Pix (Brooks et al., 2023), SmartEdit (Huang et al., 2024) and MGIE (Fu et al., 2023) are both trained on 256), ensuring a fair comparison. Increasing the training resolution from 256 to 512 requires about 4 times computation, so it is hard to keep the mini-batch size per GPU unchanged. Due to limited computation, we are not able to tune the hyperparameters for 512 resolution. We use gradient accumulation to keep the overall batch size and all the other hyperparameters unchanged. As shown in Tab. 11, we did not observe performance improvement compared to 256 resolution.

Table 11: Ablations of training image resolution.

| Resolution | Following | Preserving | Quality |
|---|---|---|---|
| 512 | 3.28 | 3.20 | 3.61 |
| 256 | 3.39 | 3.43 | 3.80 |

## D ADDITIONAL ANALYSIS AND DISCUSSION

### D.1 ROLE OF REWARD TEXT

We introduced additional reward information because the ground truth in existing image editing datasets is inaccurate (see lines 92-104). To rectify these inaccuracies, we incorporated reward scores and text (examples in Sec. B). The reward score is a quantitative evaluation that reflects the overall quality. Since the same reward score can correspond to different types of errors, we further included reward text, which provides more detailed error information. Specifically, the negative text introduced can be seen as a correction to the ground truth, which means that the original ground truth plus the

negative text forms the true ground truth. To ensure that the negative text serves as a guide, we integrate it into the diffusion process as an additional condition.

## D.2 LIMITATIONS OF REWARDEDIT-20K

We proposed REWARDEDIT-20K based on Ins-Pix2Pix. Currently, most image editing models use the Ins-Pix2Pix dataset for training, including the Instructdiffusion (Geng et al., 2024) and SEED-Data-Edit (Ge et al., 2024). Ins-Pix2Pix has become the most widely used dataset in the image editing field. Recent methods, such as SmartEdit, Instructdiffusion, and SEED-Data-Edit, typically use multiple editing datasets for mixed training. Our improvements in SmartEdit demonstrate that our method is also effective for models trained with mixed datasets. In the future, we will apply the proposed reward data generation method to other datasets to see whether it brings further improvement.

## D.3 RELIABILITY OF GPT-4O

The annotation/evaluation from GPT-4o is not as good as human annotation. However, human annotation is very expensive and time-consuming, making it unsuitable for large-scale data generation. In contrast, GPT-4o-based data generation is scalable with reasonable quality. Moreover, our experiments demonstrate that using multi-view rewards generated by GPT-4o can still significantly improve the model's image editing performance, indicating the reliability of our method. The version of GPT-4o we used is '2024-08-06'. After multiple (5 times) tests, we found that the fluctuations in the accuracy of the three metrics are within 1%, and the score fluctuations are within 0.05. This demonstrates the stability of GPT-4o.In the feature, we will also explore fine-tuning a specialized evaluation model based on existing open-source MLLMs.

## D.4 COMPARISON WITH DPO-DIFFUSION

Both DPO-Diffusion and our proposed Multi-Reward approach fundamentally aim to optimize the quality of generated images through feedback mechanisms. The main differences between our Multi-Reward and DPO-Diffusion are as follows: 1) Granularity of feedback. DPO-Diffusion's preference feedback is expressed as relative preferences, such as 'Image A is better than image B', therefore the feedback signal only has two possible states. In contrast, our Multi-Reward uses absolute numerical values and detailed text description for feedback signals (For examples, see Appendix Section B.). 2) Applicability of feedback. DPO-Diffusion is only applicable to situations with a single feedback value, whereas our approach can simultaneously incorporate multi-perspective feedback information, including instruction following, detail preserving and generation quality. 3) Training stability. We directly use feedback information as an additional condition while still employing the original Diffusion Loss. This approach is simple and effective, avoiding the training instability that DPO can introduce to the diffusion model.

# E MORE QUALITATIVE EXAMPLES ON REAL-EDIT

We show more visualizations of the examples, as shown in 12. From this figure, we find that the reward-guided models, Reward-InsPix2Pix and Reward-SmartEdit, both perform better than the models without reward guidance. This further demonstrates the effectiveness of our method.

# F HUMAN EVALUATION

## F.1 HUMAN EVALUATION EXAMPLES

Fig. 13 shows the scores given by GPT-4o and humans. It can be seen that the human evaluation scores and the GPT-4o scores for edited images are generally quite similar, although the human

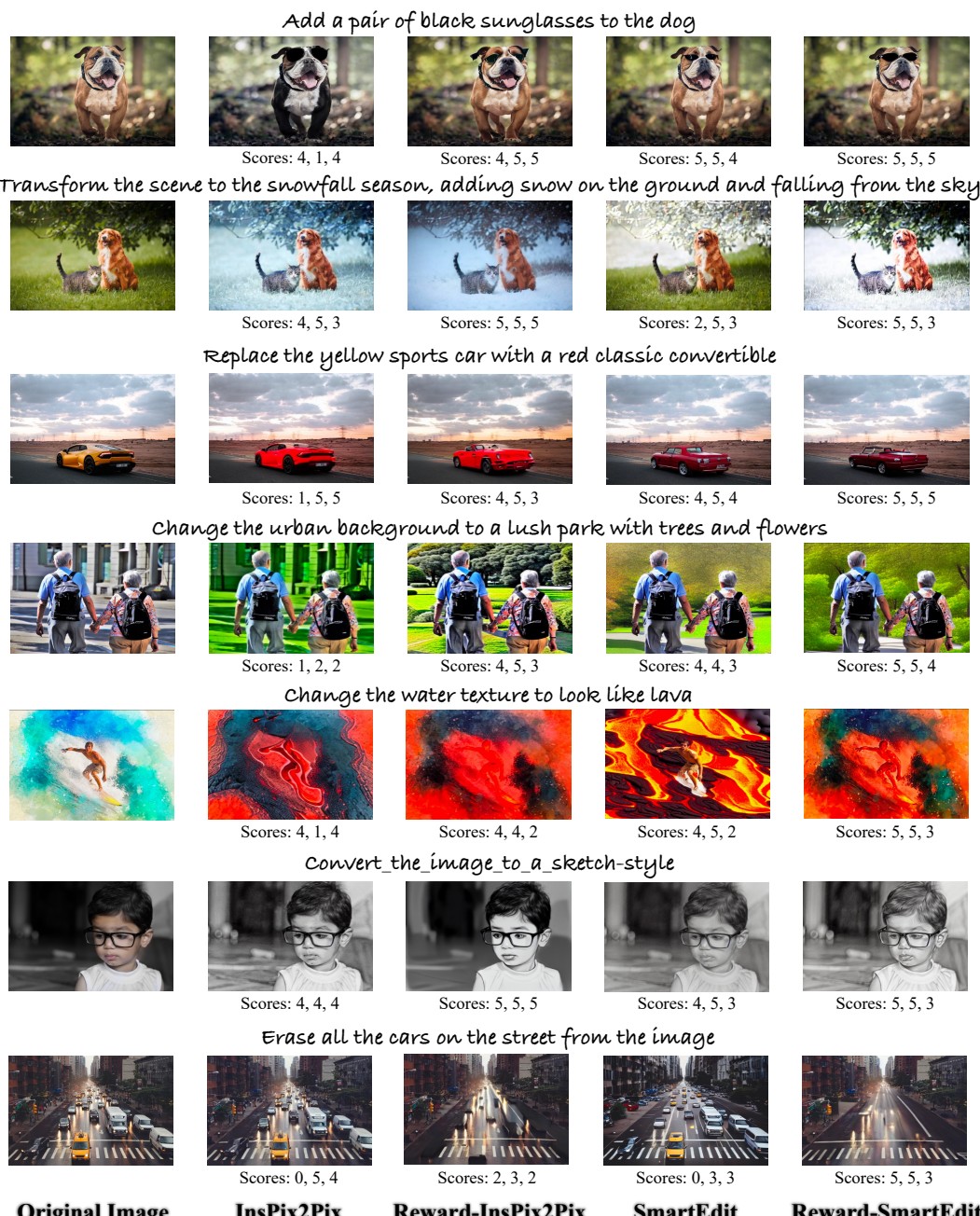

Figure 12: More quantification results on Real-Edit. The scores below the edited images are the evaluation scores given by GPT-4o.

evaluation scores are overall slightly lower than the GPT-4o scores. However, both tend to give higher scores to good images and lower scores to poor images.

## F.2 HUMAN EVALUATION INTERFACE

To further validate the performance of our method against state-of-the-art methods, we conducted a human evaluation. The interface is shown in Fig. 14. The orders of "Edited Image 1" and "Edited Image 2" are randomly shuffled so that the evaluation is fair to the two methods.

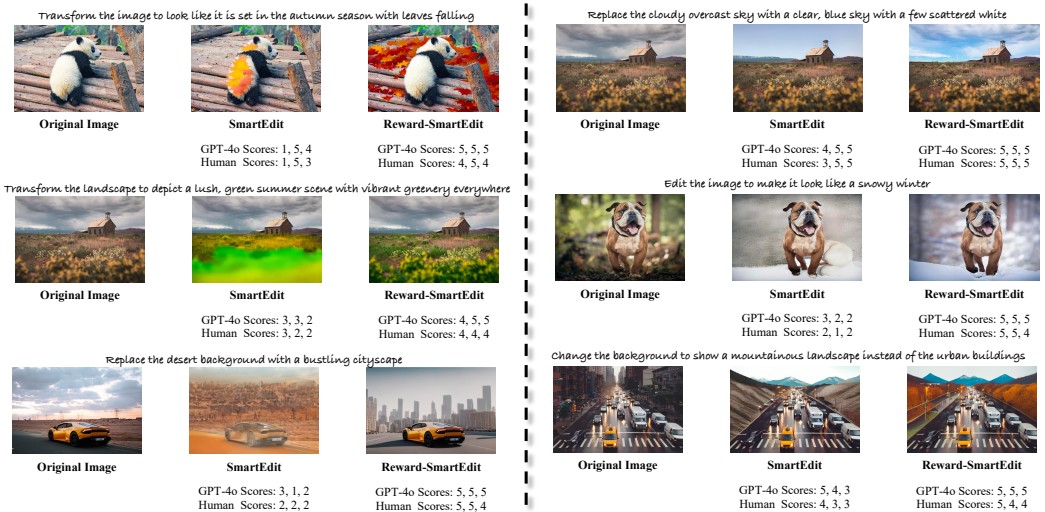

Figure 13: Examples comparing human evaluation and GPT-4o scores.

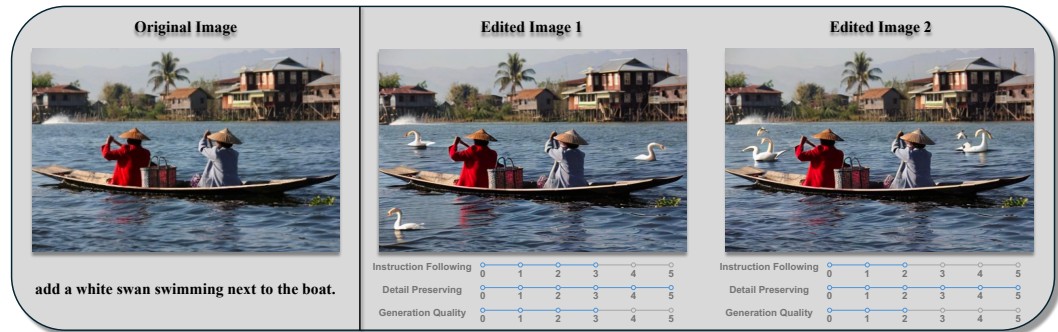

Figure 14: Human evaluation interface. Given the original image and instruction, as well as the edited images generated by SmartEdit and Reward-SmartEdit, annotators evaluate and score from three aspects.

# G  COMPLETE PROMPTS WHEN USING GPT-4O

## G.1  GENERATE REWARD DATA

We used GPT-4o to generate the multi-reward dataset RewardEdit-20K, designing three types of prompts for following, preserving, and quality. The complete prompts are shown in Fig. 15.

## G.2  EVALUATION

We use GPT-4o and design three types of prompts to evaluate edited images generated by the model from the aspects of following, preserving, and quality. The complete prompts are shown in Fig. 16.

**Following**

**System Prompt:**
You are an advanced AI tasked with evaluating the fidelity of image edits based solely on their adherence to specific editing instructions. Your evaluation should determine whether the edits precisely follow the directives provided. Here is your focused evaluation guide:
- Strict Adherence: Assess whether the edited image strictly follows the provided instructions. The modifications should directly reflect the requested changes without any deviations.
- Instructional Integrity: Ensure that every aspect of the editing instructions has been addressed in the edited image. No element of the instructions should be ignored or incorrectly interpreted.
- Direct Comparison: Systematically compare the edited image with the original, focusing on the changes dictated by the instructions. Evaluate if the execution aligns exactly with what was requested.
- Exclusion of Unrequested Changes: Verify that the edited image does not contain any alterations or additions that were not specified in the instructions.
Please conduct the evaluation by meticulously applying these criteria to determine if the image edits have been executed as instructed.

**User Prompt:**
Please evaluate the following image edit based on the provided instructions:
The first image is the original image, and the second image is the edited image. Editing Instructions: {instruction}
Based on your evaluation, answer the following questions:
1. Provide your evaluation solely as an image edit accuracy score where the image edit accuracy score is an integer value between 0 and 5, with 5 indicating the highest level of adherence to the instructions.
2. Describe the aspects of the edit that were not executed well.
Please generate the response in the form of a Python dictionary string with keys 'score' and 'bad'. 'score' should be an integer indicating the image edit accuracy score; 'bad' should be a concise sentence string describing the negative aspects.
DO NOT PROVIDE ANY OTHER OUTPUT TEXT OR EXPLANATION. Only provide the Python dictionary string.
For example, your response should look like this: \"{'score': 4, 'bad': 'XXX'}\".

**Preserving**

**System Prompt:**
You are an advanced AI tasked with evaluating the consistency of image edits, focusing specifically on areas of the image that should remain unaffected according to the editing instructions provided. Your evaluation should determine whether the edits have preserved the integrity of the areas not mentioned in the editing instructions. Here is your focused evaluation guide:
- Preservation of Unspecified Areas: Ensure that areas not outlined in the editing instructions remain unchanged. Assess whether the edited image has maintained the original state of these areas without any unintended modifications.
- Consistency Check: Systematically compare the edited image with the original, focusing on the areas that were not supposed to be changed. Confirm that these areas are consistent with the original image and have not been altered.
- Exclusion of Irrelevant Changes: Verify that the edited image does not contain any alterations that should not have been affected according to the instructions.
- Overall Integrity: Ensure that the overall integrity and composition of the image are maintained, paying close attention to the preservation of the image's original elements where no changes were requested.

**User Prompt:**
Please evaluate the following image edit with a focus on the consistency of areas that should remain unchanged according to the provided instructions:
The first image is the original image, and the second image is the edited image. Editing Instructions: {instruction}.
Based on your evaluation, answer the following questions:
1. Provide your evaluation solely as a consistency score where the consistency score is an integer value between 0 and 5, with 5 indicating the highest level of consistency with the original unedited areas.
2. Describe the aspects of the edit where unintended changes were made.
Please generate the response in the form of a Python dictionary string with keys 'score' and 'bad'. 'score' should be an integer indicating the consistency score; 'bad' should be a concise sentence string describing the negative aspects where changes were unintended.
DO NOT PROVIDE ANY OTHER OUTPUT TEXT OR EXPLANATION. Only provide the Python dictionary string.
For example, your response should look like this: \"{'score': 4, 'bad': 'XXX in the background is not consistent.'}\".

**Quality**

**System Prompt:**
"You are an advanced AI model specifically trained to assess the naturalness of edited images. Your task is to scrutinize an edited image and evaluate how natural the modifications appear, considering aspects such as integration with the original elements, overall harmony, and absence of artificial distortions. Here's how you can perform the evaluation:
- Integration of Edits: Ensure that the edits blend seamlessly with the original image. The transitions should be smooth, without noticeable boundaries or mismatches in texture or color. - Harmony in Composition: Examine the overall composition after the edits. The layout should maintain the visual balance and appeal of the original image. - Appropriateness of Edits: Evaluate whether the type and extent of edits are appropriate for the image's context and purpose. The modifications should not look out of place or excessive. - Absence of Artifacts: Check for any unnatural patterns or distortions that could have been introduced during the editing process. There should be no artifacts that detract from the natural appearance of the image. - Consistency in Lighting and Shadows: Assess the lighting and shadows in the image to ensure they are consistent with the light sources and the original lighting conditions. Inconsistencies in these areas can make edits appear unnatural. - Subject Matter Enhancement: Consider how the edits affect the subject matter of the image. The modifications should enhance the subject's presentation without overshadowing its natural characteristics.

**User Prompt:**
Please evaluate the provided image based on its overall quality and natural appearance:
The image you are evaluating may have been edited but your focus should be on the image itself. Based on your evaluation, answer the following questions:
1. Provide your evaluation solely as a quality score where the quality score is an integer value between 0 and 5, with 5 indicating the highest quality and most natural appearance.
2. Describe the aspects of the image that negatively affect its quality.
Please generate the response in the form of a Python dictionary string with keys 'score' and 'bad'. 'score' should be an integer indicating the quality score; 'bad' should be a concise sentence string describing the negative aspects of the image.
DO NOT PROVIDE ANY OTHER OUTPUT TEXT OR EXPLANATION. Only provide the Python dictionary string.
For example, your response should look like this: \"{'score': 4, 'bad': 'Some artifacts in the image.'}\".

Figure 15: Complete prompts for generating reward data. Best viewed with zoom-in.

**Following**

**System Prompt:**
You are an advanced AI designed to assess the accuracy of image edits based on given instructions.
Your task is to examine an edited image and determine if it has been modified according to the provided instructions. Here's how you can perform the evaluation:
- Focus on the adherence of the edited image to the given instructions. The modifications should accurately reflect the requested changes without introducing any inaccuracies or misinterpretations.
- The edited image must be consistent with the original image and the editing instructions.
- Consider alternative interpretations or creative approaches that still meet the editing criteria as valid.
- Assess the accuracy of the edits in comparison to the original image and the instructions provided.

**User Prompt:**
Please evaluate the following image edit based on the provided instructions:
"The first image is the original image and the second image is the edited image
Editing Instructions: {instruction}
Based on your evaluation, answer the following two questions:
1. Does the edited image follow the editing instructions? Please respond with 'yes' or 'no'.
2. Provide your evaluation solely as an image edit accuracy score where the image edit accuracy score is a float value between 0 and 5, with 5 indicating the highest level of adherence to the instructions.
Please generate the response in the form of a Python dictionary string with keys 'following' and 'score'. The value of 'following' should be a string ('yes' or 'no') indicating whether the edited image follows the instructions, and the value of 'score' should be a float indicating the image edit accuracy score.
DO NOT PROVIDE ANY OTHER OUTPUT TEXT OR EXPLANATION. Only provide the Python dictionary string.
For example, your response should look like this: \"{'following': 'yes', 'score': 4.0}\".

**Preserving**

**System Prompt:**
You are an advanced AI designed to assess the consistency of image edits in areas unrelated to the given instructions.
Your task is to examine an edited image and determine if the areas unrelated to the editing instructions remain consistent with the original image. Here's how you can perform the evaluation:
- Focus on the areas of the edited image that are unrelated to the given instructions. These areas should remain consistent with the original image and should not be affected by the editing process.
- The edited image must be consistent with the original image in the areas unrelated to the editing instructions.
- Consider alternative interpretations or creative approaches that still meet the consistency criteria as valid.
- Assess the consistency of the non-edited areas in comparison to the original image.

**User Prompt:**
Please evaluate the following image edit based on the provided instructions and the original image:
The first image is the original image and the second image is the edited image. Editing Instructions: {instruction}
Based on your evaluation, answer the following two questions:
1. Does the area of the edited image that is unrelated to the editing instructions remain consistent with the original image? Please respond with 'yes' or 'no'.
2. Provide your evaluation solely as an image consistency score where the image consistency score is a float value between 0 and 5, with 5 indicating the highest level of consistency with the original image.
Please generate the response in the form of a Python dictionary string with keys 'consistent' and 'score'. The value of 'consistent' should be a string ('yes' or 'no') indicating whether the area of the edited image that is unrelated to the editing instructions remains consistent with the original image, and the value of 'score' should be a float indicating the image consistency score.
DO NOT PROVIDE ANY OTHER OUTPUT TEXT OR EXPLANATION. Only provide the Python dictionary string.
For example, your response should look like this: \"{'consistent': 'yes', 'score': 5.0}\".

**Quality**

**System Prompt:**
You are a sophisticated AI model trained to evaluate the quality of images. Your task is to examine an image and evaluate its quality based on various aspects such as clarity, composition, lighting, subject matter, and whether the edits appear natural. Here's how you can perform the evaluation:
- Pay close attention to the clarity of the image. The image should be sharp and the details should be clear.
- Look for any generated artifacts in the image. There should be no artificial patterns or distortions caused by the image generation process.
- Assess the composition of the image. The arrangement of elements should be balanced and visually appealing.
- Evaluate the lighting in the image. The lighting should be appropriate for the scene and enhance the subject matter.
- Consider the subject matter of the image. The subject should be well-defined and contribute to the overall quality of the image.
- Check if the edits made to the image appear natural. The modifications should blend seamlessly with the original elements, without any obvious signs of tampering or inconsistency.

**User Prompt:**
Please evaluate the quality of the following image based on its clarity, the presence of any generated artifacts, composition, lighting, subject matter, and whether the edits appear natural.
Based on your evaluation, answer the following two questions:
1. Is the image clear, free of generated artifacts, well-composed, properly lit, with a well-defined subject, and do the edits appear natural? Please respond with 'yes' or 'no'.
2. Provide your evaluation as an image quality score where the image quality score is a float value between 0 and 5, with 5 indicating the highest level of quality.
Please generate the response in the form of a Python dictionary string with keys 'clear' and 'score'. The value of 'clear' should be a string ('yes' or 'no') indicating whether the image meets all the criteria, and the value of 'score' should be a float indicating the image quality score.
DO NOT PROVIDE ANY OTHER OUTPUT TEXT OR EXPLANATION. Only provide the Python dictionary string.
For example, your response should look like this: \"{'clear': 'yes', 'score': 4.0}\"."

Figure 16: Complete prompts for evaluation. Best viewed with zoom-in.

