# OpenReview forum: "Multi-Reward as Condition for Instruction-based Image Editing"
_ICLR.cc/2025/Conference — ICLR 2025 Poster_

### Official Review · Reviewer_sJ5c · 2024-11-02

**Soundness:** 3
**Presentation:** 3
**Contribution:** 2
**Rating:** 6
**Confidence:** 5

**Summary:**

This paper introduces a dataset and benchmark for assessing image editing performance across multiple dimensions. The authors also utilize these multi-dimensional scores as rewards to enhance the effectiveness of image editing models.

**Strengths:**

1. The paper underscores a valuable methodology for evaluating image editing datasets, which is highly beneficial in practical applications.
2. It proposes the use of evaluation scores to boost the performance of image editing models, which is novel in image editing.

**Weaknesses:**

1. Although the use of GPT-4o for evaluation is efficient in practice, it lacks guaranteed consistency. The evaluation results could be influenced by underlying changes in GPT-4o, rendering the results unreliable. An alternative solution to this issue could be to train an independent evaluation model.

2. The RewardEdit-20K dataset continues to utilize images generated by InstructPix2Pix's model. For some challenging editing samples, scores cannot exceed 2 points. Even if the reward model can enhance performance in other samples, can it generate better results for these difficult cases generated by InsPix2Pix?

3. There is a lack of detailed analysis on the individual impact of each perspective reward on the improvement of editing models. Including such insights in an ablation study could strengthen the paper's contribution.

4. The practice of incorporating additional "reward scores" to enhance image generation quality is already established within the stable diffusion community (see https://civitai.com/articles/4248/what-is-score9-and-how-to-use-it-in-pony-diffusion). A more thorough discussion linking the proposed multi-reward framework to existing methodologies would enrich the manuscript's contribution.

**Questions:**

1. Could you elaborate on the dimensionality of $c_R$ and $Linear(c_R)$, and explain how lines 281 and 287 can be implemented, given that both use the same notation of $Linear(c_R)$?
2. In line 369, is there a specific reason for training at a resolution of 256? Wouldn't training and inference at 512 yield better results?
3. Are the trained modules shared or trained separately between InstructPix2Pix and smartEdit?

---

> ### Author Response · Authors · 2024-11-21
> **Rebuttal (part 1/2)**
>
> We appreciate the reviewer for careful comments and provide our responses below. All changes in the revision are marked in red.
>
> > **Q1**: Although the use of GPT-4o for evaluation is efficient in practice, it lacks guaranteed consistency. The evaluation results could be influenced by underlying changes in GPT-4o, rendering the results unreliable. An alternative solution to this issue could be to train an independent evaluation model.
>
> **A1**: Thanks for this comment. The version of GPT-4o we used is '2024-08-06'. After multiple (5 times) tests, we found that the fluctuations in the accuracy of the three metrics are within **1%**, and the score fluctuations are within **0.05** on Real-Edit benchmark. This demonstrates the stability of GPT-4o. However, we fully agree with the reviewer's suggestion to train an independent evaluation model. In the future, we will explore fine-tuning a specialized evaluation model based on existing open-source multimodal large model.
>
> We have included the above discussion in *Section D.3* of the appendix in revision.
>
>
> > **Q2**: The RewardEdit-20K dataset continues to utilize images generated by InstructPix2Pix's model. For some challenging editing samples, scores cannot exceed 2 points. Even if the reward model can enhance performance in other samples, can it generate better results for these difficult cases generated by InsPix2Pix?
>
> **A2**: Thank you for your insightful comment. To investigate whether our reward model can generate better edited images for challenging editing samples in InsPix2Pix, we first randomly selected 500 samples from RewardEdit-20K with scores not exceeding 2. Then, we used our reward model to generate edited images based on the original images and instructions of these samples, and scored them using GPT-4o. The experimental results are shown in the table below. "Original" represents the average scores of the original edited images of these samples across three metrics, while "Ours" represents the scores of the edited images generated by our reward model. From the table, it can be observed that the edited images generated by our method significantly outperform the original edited images on all three metrics, indicating that our method can generate better results for these difficult cases.
>
> **Tab. C: Comparison of edited images for challenging samples in InsPix2Pix.**
>
> |   &nbsp;&nbsp;Method&nbsp;&nbsp;   | &nbsp;&nbsp;Following&nbsp;&nbsp; | &nbsp;&nbsp;Preserving&nbsp;&nbsp; | &nbsp;&nbsp;Quality&nbsp;&nbsp; |
> |:----------:|:---------:|:----------:|:-------:|
> |  Original  |    1.15   |    1.69    |   1.99  |
> |    Ours    |    2.92   |    4.10    |   3.68  |
>
>
> We have integrated the above results and analysis in *Section C.3* of the appendix in revision.
>
>
> > **Q3**: There is a lack of detailed analysis on the individual impact of each perspective reward on the improvement of editing models. Including such insights in an ablation study could strengthen the paper's contribution.
>
> **A3**: Thank you for your suggestion. We agree that analyzing the impact of each perspective reward is beneficial and we conduct additional experiments for training on each perspective separately. As shown in the table below, the following score reached 3.40 with only the instruction following reward, the preserving score reached 3.54 with only the detail preserving reward, and the quality score reached 3.95 with only the generation quality reward. These results demonstrate the effectiveness of each perspective reward.
>
>
> **Tab. D: Ablation of each perspective reward. 'IF', 'DP' and 'GQ' are instruction following reward, detail preserving reward and generation quality reward.**
>
> | &nbsp;&nbsp; IF &nbsp;&nbsp; | &nbsp;&nbsp; DP &nbsp;&nbsp; | &nbsp;&nbsp; GQ &nbsp;&nbsp; | &nbsp;&nbsp;Following &nbsp;&nbsp; | &nbsp;&nbsp;Preserving&nbsp;&nbsp; | &nbsp;&nbsp;Quality&nbsp;&nbsp; |
> |:----:|:----:|:----:|:---------:|:----------:|:-------:|
> |  ✓   |      |      |    3.40   |    3.25    |   3.72  |
> |      |  ✓   |      |    3.23   |    3.54    |   4.00  |
> |      |      |  ✓   |    3.20   |    3.23    |   3.95  |
> |  ✓   |  ✓   |  ✓   |    3.39   |    3.43    |   3.80  |
>
>
> We have included the above results in *Section C.4* of the appendix in revision to make our contributions clearer.

---

> > ### Author Response · Authors · 2024-11-21
> > **Rebuttal (part 2/2)**
> >
> > > **Q4**: The practice of incorporating additional "reward scores" to enhance image generation quality is already established within the stable diffusion community (see https://civitai.com/articles/4248/what-is-score9-and-how-to-use-it-in-pony-diffusion). A more thorough discussion linking the proposed multi-reward framework to existing methodologies would enrich the manuscript's contribution.
> >
> > **A4**: Thanks for this comment. We discussed our method in relation to existing reward-based methods in *Section 2.2* of the Related Work. In the Text-to-Image, some work has explored the use of reward scores to enhance the quality of generated images. For example, Pony Diffusion, as mentioned by the reviewer, employs a CLIP-based aesthetic ranking method to generate reward scores. Different from these works, our reward information comes from GPT-4o, which includes not only reward scores but also reward text. In addition, we focus more on the role of reward information in the image editing domain rather than text-to-image generation.
> >
> > We have added the discussions to  the Related Work of the revision.
> >
> >
> > > **Q5**: Could you elaborate on the dimensionality of $c_R$ and $Linear(c_R)$, and explain how lines 281 and 287 can be implemented, given that both use the same notation of $Linear(c_R)$?
> >
> > **A5**: Thanks for this comment. The reward condition $c_R$ has a dimension of 768. In the Reward Encoder module, to match the dimension of latent noise, a linear layer (line 281) is used to transform the dimension of $c_R$ to 320. In the Unet module, the dimension is 1280, so a linear layer (line 287) is used to transform the dimension of $c_R$  to 1280. We have clarified this point in revision.
> >
> >
> > > **Q6**: In line 369, is there a specific reason for training at a resolution of 256? Wouldn't training and inference at 512 yield better results?
> >
> > **A6**: Thanks for this comment. We chose to train at a resolution of 256 to maintain consistency with other methods (InsPix2Pix, SmartEdit, MGIE and HQ-Edit are both trained on 256), ensuring a **fair comparison**. Increasing the training resolution from 256 to 512 requires about 4 times computation, so it is hard to keep the mini-batch size per GPU unchanged. Due to limited computation, we are not able to tune the hyperparameters for 512 resolution. We use gradient accumulation to keep the overall batch size and all the other hyperparameters unchanged. We did not observe performance improvement compared to 256 resolution.
> >
> > **Tab. E: Ablations of training image resolution.**
> > |  &nbsp; &nbsp;Resolution &nbsp; &nbsp; |  &nbsp; &nbsp;Following &nbsp; &nbsp; |  &nbsp; &nbsp;Preserving &nbsp; &nbsp; |  &nbsp; &nbsp;Quality &nbsp; &nbsp; |
> > |:----------:|:---------:|:----------:|:-------:|
> > |    512     |    3.28   |    3.20    |   3.61  |
> > |    256     |    3.39   |    3.43    |   3.80  |
> >
> >
> > We have added this point and results in *Section C.5* of the appendix in revision.
> >
> >
> > > **Q7**: Are the trained modules shared or trained separately between InstructPix2Pix and smartEdit?
> >
> > **A7**: Thanks for this comment. The proposed MRC module is trained separately for InstructPix2Pix and smartEdit without sharing weights. We have clarified this point in Line 403 in revision.

---

> ### Comment · Reviewer_sJ5c · 2024-11-25
>
> Thank you for your rebuttal. However, there are still some concerns that need to be addressed.
> ## About A3
> The ablation study in A3 is lacking in certain aspects. Specifically, I have two questions:
> 1. The three evaluation metrics for each individual reward seem similar. Please analyze why the metrics increase even when the reward is not related to the specific metric.
> 2. Why is the evaluation of the full model inferior to that obtained by adding individual rewards?
> ## About A5
> There appears to be a bug between Equation line 281 and Figure 5. From line 281, the latent behaves as query only, then the resulting output of the cross attention $Z''$ eliminates the detailed information of the source image.  However in Figure 5,  $Z''$ is directly sent to the diffusion model. Per my understanding, the structure of Figure 5 is not feasible, as the diffusion model can not touch the details of the input source image, making it unable to perform image editing.
>
> # Suggestion
> It is highly recommended to label each equation with a numerical label, as this is a standard practice.
>
> # Summary
> I am inclined to reject this paper as the readability could be significantly enhanced.

---

> > ### Author Response · Authors · 2024-11-26
> >
> > Thanks for the helpful comments, we address the concerns in the following items.
> >
> > > **About A3**: (1) The three evaluation metrics for each individual reward seem similar. Please analyze why the metrics increase even when the reward is not related to the specific metric. (2) Why is the evaluation of the full model inferior to that obtained by adding individual rewards?
> >
> > **For (1)**, the three types of rewards come from three different perspectives. Although these perspectives are independent by definition, they essentially aim to *rectify the inaccurate supervision and therefore influence each other*. For example, in the case shown in Fig. 1 (b) with the instruction "make the glasses green", the ground-truth edited image incorrectly changes the background and clothes to green as well. This could mislead the model into thinking that "make the glasses green" requires changing the background and clothes to green, resulting in incorrect instruction following. However, if a detail preserving reward is added, indicating that "the colors of clothes and background are not consistent", it helps the model correctly understand the instruction, thereby indirectly helping with instruction following. Therefore, specific rewards not only improve the corresponding metrics but also enhance the other two metrics.
> >
> > **For (2)**, from Tab. D, we see that using all three rewards simultaneously does not achieve SOTA for each metric. This is due to some negative interactions among the three perspectives; for example, strong instruction following might reduce detail preserving, and strong detail preserving might inhibit instruction following. Since editing is the core task and the sample is considered failed if the image is not edited at all, we prioritize the following score and then compare preserving and quality when following scores are similar. From Tab. D, when using all three rewards, the following score achieves 3.39, while preserving and quality achieve 3.43 and 3.80, respectively, demonstrating the effectiveness of using the three rewards together.
> >
> > **Tab. D: Ablation of each perspective reward. 'IF', 'DP' and 'GQ' are instruction following, detail preserving and generation quality reward.**
> >
> > | &nbsp;&nbsp; IF&nbsp;&nbsp;  |  &nbsp;&nbsp;DP&nbsp;&nbsp;  | &nbsp;&nbsp; GQ &nbsp;&nbsp; | &nbsp;&nbsp;Following&nbsp;&nbsp; | &nbsp;&nbsp;Preserving&nbsp;&nbsp; | &nbsp;&nbsp;Quality&nbsp;&nbsp; |
> > |:----:|:----:|:----:|:---------:|:----------:|:-------:|
> > |  ✓   |      |      |    3.40   |    3.25    |   3.72  |
> > |      |  ✓   |      |    3.23   |    3.54    |   4.00  |
> > |      |      |  ✓   |    3.20   |    3.23    |   3.95  |
> > |  ✓   |  ✓   |  ✓   |    3.39   |    3.43    |   3.80  |
> >
> >
> > > **About A5**: There appears to be a bug between Equation line 281 and Figure 5. From line 281, the latent behaves as query only, then the resulting output of the cross attention $Z''$ eliminates the detailed information of the source image. However in Figure 5, $Z''$ is directly sent to the diffusion model. Per my understanding, the structure of Figure 5 is not feasible, as the diffusion model can not touch the details of the input source image, making it unable to perform image editing.
> >
> > [A] Vaswani, A. "Attention is all you need." Advances in Neural Information Processing Systems (2017).
> >
> > Sorry for the confusion. The $Z_t'$ is obtained by concatenating $Z_t$ with original image condition $c_I$ and fusing them through convolution, thus it contains details of the input source image. For line 281, we use latent noise $Z'_t$ as the query of the reward encoder, which is a standard transformer encoder block from [A]. As shown in Fig. 12 in the paper, it contains a skip connection so that the output is initialized as the input $Z'_t$ and the block only learns residual information if necessary. Therefore, the $Z''$ still contains the detail information of the input source image as well as the reward information. We will make this clear in the final version.
> >
> >
> > > **Others**
> >
> > Thanks to the reviewer's suggestion, we have already labeled each equation with a numerical label. We will work on revising the manuscript repeatedly to improve the readability of the paper.

---

> > > ### Comment · Reviewer_sJ5c · 2024-11-26
> > >
> > > Thanks for the authors' follow-up response. The response resolves my concerns, and the revised version is now more readable. Given this, I will raise my rating based on the revised version.

---

> > > > ### Author Response · Authors · 2024-11-26
> > > >
> > > > We are happy to hear that our response addressed your concerns. We sincerely appreciate the time and effort you have dedicated to reviewing our work.

---

### Official Review · Reviewer_eTqX · 2024-11-03

**Soundness:** 3
**Presentation:** 3
**Contribution:** 4
**Rating:** 6
**Confidence:** 3

**Summary:**

This paper identifies significant issues related to data quality in the current Image Edit dataset. It proposes a pipeline for data cleaning and scoring using MLLM and introduces a cleaned dataset along with a training framework designed for multi-reward scenarios.

**Strengths:**

- This paper discusses the issues present in the current dataset and utilizes MLLM for data cleaning. Through experiments and comparisons, it demonstrates the effectiveness of the cleaned dataset and the new training methods.
- Good writing and detailed experiments make this paper compelling.

**Weaknesses:**

- W.1: The paper lacks a comparison with RL in T2I methods like DPO-Diffusion[1]. If I understand correctly, I believe that the Multi-Reward Framework is conceptually similar to methods like DPO-Diffusion. Therefore, I think it is reasonable and necessary to articulate the comparisons and distinctions between these approaches, especially the method difference.
- W.2: The article lacks some novelty. Of course, high-quality and abundant data can effectively enhance model performance, so I am uncertain about the extent to which the proposed Multi-Reward Framework improves upon traditional methods. For example, in Table 1, additional editing data (0.02M) was used for training. I believe it is fair to compare it with the exact same data using the same baseline method; otherwise, it is difficult to convince me whether the improvement comes from the high-quality data or the architecture. Perhaps experiments on the unprocessed REWARDEDIT-20K data could be added for comparison.

[1] Wallace, B., Dang, M., Rafailov, R., Zhou, L., Lou, A., Purushwalkam, S., Ermon, S., Xiong, C., Joty, S., & Naik, N. (2023). *Diffusion Model Alignment Using Direct Preference Optimization*. arXiv preprint arXiv:2311.12908.

**Questions:**

See above, especially W.1

---

> ### Author Response · Authors · 2024-11-21
> **Rebuttal**
>
> We thank the reviewer for careful comments on our work and provide our responses below. All changes in the revision are marked in red.
>
> > **Q1**: The paper lacks a comparison with RL in T2I methods like DPO-Diffusion[1]. If I understand correctly, I believe that the Multi-Reward Framework is conceptually similar to methods like DPO-Diffusion. Therefore, I think it is reasonable and necessary to articulate the comparisons and distinctions between these approaches, especially the method difference.
> [1] Wallace, B., Dang, M., Rafailov, R., Zhou, L., Lou, A., Purushwalkam, S., Ermon, S., Xiong, C., Joty, S., \& Naik, N. (2023). Diffusion Model Alignment Using Direct Preference Optimization. arXiv preprint arXiv:2311.12908.
>
> **A1**: Thanks for the insightful comment. We agree with the reviewer's suggestion that we should clarify the differences and connections between our method and methods like DPO-Diffusion. Both DPO-Diffusion and our proposed Multi-Reward approach fundamentally aim to optimize the quality of generated images through **feedback mechanisms**. The main differences between our Multi-Reward and DPO-Diffusion are as follows:
> **(1)** Granularity of feedback. DPO-Diffusion's preference feedback is expressed as relative preferences, such as `Image A is better than image B',  therefore the feedback signal only has two possible states. In contrast, our Multi-Reward uses absolute numerical values and detailed text description for feedback signals (For examples, see Appendix *Section B*).
> **(2)** Applicability of feedback. DPO-Diffusion is only applicable to situations with a single feedback value, whereas our approach can simultaneously incorporate multi-perspective feedback information, including instruction following, detail preserving and generation quality.
> **(3)** Training stability. We directly use feedback information as an additional condition while still employing the original Diffusion Loss. This approach is simple and effective, avoiding the training instability that DPO can introduce to the diffusion model.
>
> We have integrated the above detailed comparison into *Section D.4* of the appendix in the revision.
>
>
> > **Q2**: The article lacks some novelty. Of course, high-quality and abundant data can effectively enhance model performance, so I am uncertain about the extent to which the proposed Multi-Reward Framework improves upon traditional methods. For example, in Table 1, additional editing data (0.02M) was used for training. I believe it is fair to compare it with the exact same data using the same baseline method; otherwise, it is difficult to convince me whether the improvement comes from the high-quality data or the architecture. Perhaps experiments on the unprocessed REWARDEDIT-20K data could be added for comparison.
>
> **A2**: Thank you for your thoughtful feedback. **To clarify**, we are not working on using MLLMs to filter high-quality data from InsPix2Pix. The 20K samples in our RewardEdit-20K dataset are *randomly sampled* from InsPix2Pix. Our motivation is that constructing a perfect image editing dataset is challenging, and the ground truth in existing image editing datasets often contains issues. Therefore, we propose using multi-perspective rewards to **rectify the inaccurate supervision**. To more fairly demonstrate the role of multi-perspective rewards, we conducted the ablation experiments shown in the table below. The experimental results indicate that, with the same data, using multi-perspective rewards significantly improves performance compared to the baseline, demonstrating the effectiveness of multi-perspective rewards.
>
> **Tab. B: Ablation study of editing data.**
> |  &nbsp; Architecture&nbsp;  | &nbsp; Edit Data &nbsp; |  &nbsp; Following &nbsp;| &nbsp;Preserving&nbsp; | &nbsp;Quality&nbsp; |
> |:----------:|:---------:|:---------:|:----------:|:-------:|
> |  Baseline  |  0.30M    |    2.77   |    2.59    |   3.15  |
> |  Baseline |  0.32M    |    2.90   |    2.88    |   3.52  |
> |    Ours    |  0.32M    |    3.39   |    3.43    |   3.80  |
>
> We have included the above clarification and results in *Section C.2* of the appendix in the revision to make our contributions clearer.

---

> > ### Author Response · Authors · 2024-11-26
> >
> > Dear Reviewer,
> >
> > We sincerely appreciate the time and effort you have dedicated to reviewing our work. Your insightful comments and constructive feedback are highly valued.

---

> > > ### Comment · Reviewer_eTqX · 2024-11-26
> > >
> > > Thank you for your reply, I will raise my rate.

---

### Official Review · Reviewer_X16E · 2024-11-04

**Soundness:** 3
**Presentation:** 3
**Contribution:** 3
**Rating:** 6
**Confidence:** 4

**Summary:**

This paper aims to correct the noise supervision in instruction-based image editing models by using multi-view reward data as an additional condition. To achieve this, the authors collected a dataset named RewardEdit-20K, which contains 20,000 instances of multi-view reward data.

**Strengths:**

1. This paper intorduce  a multi-view reward mechanism, instead of directly improving the quality of ground-truth images, the authors utilized GPT-4o to evaluate the training data from three key perspectives: instruction adherence, detail preservation, and generation quality.

2.The RewardEdit-20K dataset and the Real-Edit evaluation benchmark.

**Weaknesses:**

1. The multi-view reward mechanism used in this paper relies entirely on GPT-4o’s evaluation. Although GPT-4o demonstrates strong capabilities in understanding and generating natural language, it may not fully capture the subtle nuances of human perception regarding image editing quality.

2. The cost for this reward is expensive as it using GPT4-o .

**Questions:**

1. The human evaluations showed slightly lower scores than those generated by GPT-4o, though both showed consistency in ranking. Could the authors provide insights into why this discrepancy exists?

2.Given the reliance on the RewardEdit-20K dataset, a more detailed release plan for this dataset (and possibly pretrained models) could be helpful.

3. While the approach shows significant improvements, an analysis on instances where the multi-reward mechanism fails or provides subpar results would be beneficial. Understanding the limitations could offer insights for future iterations or refinements of the method.

---

> ### Author Response · Authors · 2024-11-21
> **Rebuttal**
>
> We thank the reviewer for his careful and helpful comments on our work. We provide our responses below to answer the reviewer's questions. All changes in the revision are marked in red.
>
> > **Q1**: The multi-view reward mechanism used in this paper relies entirely on GPT-4o’s evaluation. Although GPT-4o demonstrates strong capabilities in understanding and generating natural language, it may not fully capture the subtle nuances of human perception regarding image editing quality.
>
> **A1**: Thank you for this comment. We agree that the annotation/evaluation from GPT-4o is not as good as human annotation. However, human annotation is very expensive and time-consuming, making it unsuitable for large-scale data generation. In contrast, GPT-4o-based data generation is scalable with reasonable quality. Moreover, our experiments demonstrate that using multi-view rewards generated by GPT-4o can still significantly improve the model's image editing performance, indicating the reliability of our method.
>
> We have included the above discussion in *Section D.3* of the appendix for clarification.
>
>
> > **Q2**: The cost for this reward is expensive as it using GPT-4o .
>
> **A2**: Thanks for this comment. The price for GPT-4o is \\$2.5 per 1 million tokens. Evaluating a single sample costs \\$0.004, and evaluating the entire Real-Edit benchmark costs approximately \\$2.24, which is acceptable/negligible compared to training costs such as GPU expenses. Frankly, using GPT-4o incurs additional costs compared to free metrics, but we believe it's worth it. As the most powerful multimodal model, GPT-4o provides more accurate evaluations of editing performance. In the future, we will attempt to train specialized editing evaluation models to replace GPT-4o.
>
>
> > **Q3**: The human evaluations showed slightly lower scores than those generated by GPT-4o, though both showed consistency in ranking. Could the authors provide insights into why this discrepancy exists?
>
> **A3**: Thanks for this comment. We guess that the reason for this discrepancy may be that human evaluators often have higher expectations (e.g., setting a higher bar for a perfect score of 5) and subjective perceptions, making them more critical of details and quality. This results in lower scores compared to GPT's evaluations.
>
> We have added the above analysis in *Section 6.3* in revision.
>
> > **Q4**: Given the reliance on the RewardEdit-20K dataset, a more detailed release plan for this dataset (and possibly pretrained models) could be helpful.
>
> **A4**: Thanks for this comment.  We are currently organizing the data and code and plan to publicly release all code (including training, testing, evaluation code) and pretrained models on GitHub within the next two months. Additionally, the multi-perspective reward dataset RewardEdit-20K and the evaluation benchmark Real-Edit will also be made available on Huggingface.
>
>
> > **Q5**: While the approach shows significant improvements, an analysis on instances where the multi-reward mechanism fails or provides subpar results would be beneficial. Understanding the limitations could offer insights for future iterations or refinements of the method.
>
> **A5**: Thank you for your suggestions. We agree that adding an analysis of the failed cases would be beneficial for our method. To explore the limitations of our method, we collected and analyzed failed cases (please see *Section A* of the appendix in the revision). The analysis revealed two main limitations of our method. The first limitation is that during testing, even when the given multi-perspective reward scores are all 5, the generated edited image does not always achieve a score of 5. This indicates that the reward information does not always perfectly guide the model, especially in some complex cases. The second limitation is that our method has difficulty accurately understanding the quantifiers and spatial position words in the instructions, as shown in Fig. 10 in revision. This may be due to the model's insufficient understanding of fine-grained language features. In future work, we will explore ways to improve the model's understanding of fine-grained semantics for image editing.
>
> We have added the above analysis to *Section A* of the appendix in the revision.

---

> > ### Comment · Reviewer_X16E · 2024-11-25
> >
> > Thank you for your rebuttal, most of the doubts have been explained. I'll maintain my score for acceptance.

---

> > > ### Author Response · Authors · 2024-11-26
> > >
> > > Dear Reviewer,
> > > Thank you once again for your time and effort in reviewing our work. We greatly appreciate the thoughtful comments and constructive feedback you have provided.

---

### Official Review · Reviewer_RGo1 · 2024-11-04

**Soundness:** 3
**Presentation:** 3
**Contribution:** 3
**Rating:** 6
**Confidence:** 4

**Summary:**

The authors introduce a high-quality editing reward dataset to address the problem of editing effectiveness due to training data quality issues with instruction-based image editing methods. This dataset contains assessment scores and reward texts, which are introduced as additional conditions to the instruction-based editing model to improve the model's ability. The authors also introduce an evaluation set to assess the quality of the instruction-based image editing model from multiple dimensions. Qualitatively and quantitatively, it is demonstrated that the present method effectively improves the quality of the instructional editing model.

**Strengths:**

1.	A novel reward-based instruction editing framework that introduces evaluation scores for VLLM as well as rewarding textual feedback to improve the capability of instruction editing models.
2.	The creation of Real-Edit provides a standardized approach to evaluate instructional editing methods in different scenarios.
3.	The method shows superior performance in both quantitative metrics and qualitative results, indicating robust editing capabilities.

**Weaknesses:**

1. Both the quantitative assessment in Table 1 and the training strategy are based on the same scoring strategy for the GPT-4o, making the evaluation of the results overly dependent on the a priori of the VLLM and making it difficult to objectively validate the strengths of this method. One solution to this dilemma is to use other assessment metrics (e.g., CLIP scores) on the three dimensions to be adopted for comparison with other methods.
2. While experiments have shown that the introduction of additional reward text can improve image editing, unfortunately, there is no analysis of why introducing negative text in this way could help guide the diffusion process towards more effective editing.
3. Are there limitations to the introduction of the dataset REWARDEDIT-20K. Different instructional editing models may have been obtained by tuning on different training sets [1][2]. While this paper achieved an advantage on Ins-Pix2Pix and its improved version SmartEdit, is it still desirable to train other models using REWARDEDIT-20K obtained from the Ins-Pix2Pix training set?
[1] Instructdiffusion: A generalist modeling interface for vision tasks, CVPR 2024
[2] SEED-Data-Edit Technical Report: A Hybrid Dataset for Instructional Image Editing, arXiv 2024

**Questions:**

Please refer to the weaknesses.

---

> ### Author Response · Authors · 2024-11-21
> **Rebuttal (part 1/2)**
>
> We thank the reviewer for helpful comments on our work and provide our responses to the reviewer's questions below. All changes in the revision are marked in red.
>
> > **Q1**: Both the quantitative assessment in Table 1 and the training strategy are based on the same scoring strategy for the GPT-4o, making the evaluation of the results overly dependent on the a priori of the VLLM and making it difficult to objectively validate the strengths of this method. One solution to this dilemma is to use other assessment metrics (e.g., CLIP scores) on the three dimensions to be adopted for comparison with other methods.
>
> **A1**: Thanks for this comment. Evaluating edited images presents significant challenges due to the diversity and uncertainty of possible edited images for a given original image and instruction. GPT-4o is currently the most powerful multi-modal understanding model, capable of accurately parsing editing instructions and comprehensively understanding the content of the original images and the editing requirements. Therefore, we use GPT-4o for automated image editing evaluation. To verify the reliability of GPT-4o, we also performed human evaluations and obtained consistent results.
>
> Additionally, we agree with the reviewers that it is necessary to evaluate the dimensions of Following,Preserving, and Quality based on existing evaluation metrics. We recalculated the performance of existing methods and our method on Real-Edit based on the CLIP score and the FID score, with the results shown in the table below. Specifically, the CLIP feature similarity between the edited image and the instruction is the following score, the similarity between the original and edited images is the preserving score, and the FID between the original and edited images is the quality score. The table shows that our method still achieved promising results and improvements over the baseline.
>
> **Tab. A: Comparison of different methods based on existing evaluation metrics.**
>
> |       Method       | &nbsp;&nbsp; Following (CLIP) &nbsp;&nbsp; | &nbsp;&nbsp;Preserving (CLIP)&nbsp;&nbsp; | &nbsp;&nbsp;Quality (FID) ↓ &nbsp;&nbsp;|
> |------------------|:----------------:|:-----------------:|:---------------:|
> |     KOSMOS-G       |       26.8       |        86.4       |       3.01      |
> |    MagicBrush      |       25.2       |        91.9       |       2.86      |
> |       MGIE         |       26.4       |        87.0       |       3.09      |
> | InstructDiffusion  |       26.3       |        86.4       |       2.89      |
> |       HIVE         |       26.3       |        89.0       |       3.08      |
> |      HQ-Edit       |       28.5       |        77.2       |       3.59      |
> |    InsPix2Pix      |       27.0       |        82.3       |       3.51      |
> | Reward-InsPix2Pix (Ours)  |       27.5       |        83.8       |       3.31      |
> |     SmartEdit      |       26.5       |        87.7       |       2.80      |
> | Reward-SmartEdit (Ours)   |       26.9       |        90.0       |       2.77      |
>
> However, these metrics also have limitations: **1)** When the editing instruction and the images are complicated, CLIP/FID score can not accurately represent the following/preserving/quality of the edited image, e.g., CLIP can not distinguish left/right. **2)** the range of the following score and preserving score is relatively small, which may make it hard to distinguish performance differences between methods.
>
> We have included the above discussion and results in *Section C.1* of the appendix in the revision.

---

> > ### Author Response · Authors · 2024-11-21
> > **Rebuttal (part 2/2)**
> >
> > > **Q2**: While experiments have shown that the introduction of additional reward text can improve image editing, unfortunately, there is no analysis of why introducing negative text in this way could help guide the diffusion process towards more effective editing.
> >
> > **A2**: Thanks for this comment. We introduced additional reward information because the ground truth in existing image editing datasets is inaccurate (see lines 92-104). To rectify these inaccuracies, we incorporated reward scores and text (examples in *Section B* of the Appendix ). The reward score is a quantitative evaluation reflecting the overall quality. Since the same reward score can correspond to different types of errors, we further included reward text, which provides more detailed error information. Specifically, the negative text introduced can be seen as a **correction** to the ground truth, which means that the original ground truth plus the negative text forms the true ground truth. To ensure that the negative text serves as a guide, we integrate it into the diffusion process as an additional condition.
> >
> > We have included the above analysis and discussion in *Section D.1* of the appendix in revision.
> >
> >
> > > **Q3**: Are there limitations to the introduction of the dataset REWARDEDIT-20K. Different instructional editing models may have been obtained by tuning on different training sets [1][2]. While this paper achieved an advantage on Ins-Pix2Pix and its improved version SmartEdit, is it still desirable to train other models using REWARDEDIT-20K obtained from the Ins-Pix2Pix training set?
> > [1] Instructdiffusion: A generalist modeling interface for vision tasks, CVPR 2024
> > [2] SEED-Data-Edit Technical Report: A Hybrid Dataset for Instructional Image Editing, arXiv 2024
> >
> > **A3**: Thank you for your thoughtful feedback. We proposed REWARDEDIT-20K based on Ins-Pix2Pix. Currently, most image editing models use the Ins-Pix2Pix dataset for training, including the Instructdiffusion[1] and SEED-Data-Edit[2] mentioned by the reviewers. Ins-Pix2Pix has become the most widely used dataset in the image editing field. Recent methods, such as SmartEdit, Instructdiffusion, and SEED-Data-Edit, typically use multiple editing datasets for mixed training. Our improvements in SmartEdit demonstrate that our method is also effective for models trained with mixed datasets. In the future, we will apply the proposed reward data generation method to other datasets to see whether it brings further improvement.
> >
> > We have added the above discussion to *Section D.2* of the appendix in the revision.

---

### Author Response · Authors · 2024-11-25
**General Rebuttal**

We appreciate all the reviewers for their thoughtful and careful feedback.
In this paper, we propose a new *comprehensive solution* to address the limitations in existing image editing, including new training data, network architecture, evaluation benchmarks, and evaluation metrics. Extensive experiments demonstrate that our proposed method can be combined with existing editing models, resulting in significant performance improvements and achieving state-of-the-art results in both GPT-4o and human evaluations.

As suggested by the reviewers, we have thoroughly revised our manuscript and address each of the issues raised in the reviews:

- Reviewer `RGo1`: We have added results on other evaluation metrics (*Q1*), further analyzed the reasons for improvements brought by negative text (*Q2*) and discussed the generalizability of the RewardEdit-20K dataset (*Q3*).

- Reviewer `X16E`: We compared and discussed the GPT-4o annotations and human annotations (*Q1*), further analyzed the costs of GPT-4o (*Q2*), the score discrepancies between GPT-4o and human evaluations (*Q3*), provided a detailed release plan (*Q4*), and added failure case analysis (*Q5*).

- Reviewer `eTqX`: We thoroughly discussed the connections and differences between our approach and methods like DPO-diffusion (*Q1*), and added ablation experiments to further demonstrate the effectiveness of our method (*Q2*).

- Reviewer `sJ5c`: We discussed the stability of GPT-4o (*Q1*), validated our method's improvements on challenging editing samples (*Q2*), added ablation experiments and analysis for each reward perspective (*Q3*),  compared it with existing reward-based methods(*Q4*), clarified the dimensionality of the linear layer (*Q5*), explained why we train at a resolution of 256 (*Q6*), and clarified that the MRC module is trained separately for different methods (*Q7*).

We once again express our heartfelt gratitude to all the reviewers for their valuable feedback, and we hope that our responses satisfactorily address all concerns. Please feel free to let us know if you have any remaining concerns and we are happy to address them!

---

### Meta-Review · Area_Chair_GcaX · 2024-12-17

**Metareview:**

This paper aims to use multi-view reward data as an additional condition to address the problem of editing effectiveness caused by the issues of training data quality. To achieve this, the authors introduce a high-quality editing reward dataset and propose a benchmark to evaluate the quality of the instruction-based image editing model from multiple dimensions. Qualitative and quantitative evaluations are presented and demonstrate the effectiveness of the proposed method. All reviewers give positive rating scores. Based on the above considerations, I recommend to accept this manuscript.

**Additional Comments On Reviewer Discussion:**

The authors provided rebuttals for each reviewer, and most of reviewer present responses. During the rebuttal period, Reviewer eTqX and Reviewer sJ5c raise their rating scores considering that their problems are addressed. Reviewer RGo1 and Reviewer X16E keep their positive rating scores.

---

### Decision · Program_Chairs · 2025-01-22

Accept (Poster)